# Longitudinal associations of an exposome score with serum metabolites from childhood to adolescence
Darren R. Healy [1] ✉, Iman Zarei[1], Santtu Mikkonen [2,3], Sonja Soininen [4,5], Anna Viitasalo[4], Eero A. Haapala[4,6], Seppo Auriola[7,8], Kati Hanhineva[1,9], Marjukka Kolehmainen [1,12] & Timo A. Lakka [4,10,11,12]

Environmental and lifestyle factors, including air pollution, impaired diet, and low physical activity, have been associated with cardiometabolic risk factors in childhood and adolescence. However, environmental and lifestyle exposures do not exert their physiological effects in isolation. This study investigated associations between an exposome score to measure the impact of multiple exposures, including diet, physical activity, sleep duration, air pollution, and socioeconomic status, and serum metabolites measured using LC-MS and NMR, compared to the individual components of the score. A general population of 504 children aged 6–9 years at baseline was followed up for eight years. Data were analysed with linear mixed-effects models using the R software. The exposome score was associated with 31 metabolites, of which 12 metabolites were not associated with any individual exposure category. These findings highlight the value of a composite score to predict metabolic changes associated with multiple environmental and lifestyle exposures since childhood.

Environmental and lifestyle factors, such as air pollution, unhealthy diet and lower levels physical activity, have been associated with cardiometabolic risk factors in childhood and adolescence[1–6]. However, environmental and lifestyle exposures do not exert their physiological effects in isolation. Instead, there is complex interplay between external exposures, and with their associated internal physiological responses[7–11]. Thus, there may be a need to explore and evaluate the combined impact of environmental and lifestyle exposures on cardiometabolic health since childhood, as opposed to just assessing individual exposures, to increase our understanding of their ability to predict cardiometabolic risk. This may help to identify optimal environmental and lifestyle conditions to prevent cardiometabolic diseases since childhood.

The underlying pathophysiological processes for cardiometabolic risk originate in early-life[12], and often track from childhood to adulthood[13]. The concept of an "exposome" was proposed in 2005 and represents life-course environmental exposures[14], which provides a useful framework to assess the

development of cardiometabolic diseases since childhood. However, measuring the exposome can be challenging[15]; for example, it can be difficult to account for the sheer number of exposures individuals are exposed to over their life course, the different types of exposures and the tools/technologies able to measure these exposures, and the ability to accurately capture an individual's personal exposure[15,16]. One approach used in other areas of research that could be adopted for exposome research is the use of scores. Scores have been used in health sciences to assess cardiometabolic health, with the cardiometabolic risk score for children and adolescents[17] and "Life's Essential 8" proposed by the American Heart Association[18]. Sum scores have also been used to assess combined lifestyle factors and their impact on health[19], however, the majority of such studies have been carried out in adults. Along with measuring multiple aspects of health, sum scores have also been used to measure diet quality, where the overall impact of the diet is assessed as opposed to investigating single nutrients. Examples of the use of

[1]Institute of Public Health and Clinical Nutrition, University of Eastern Finland, Kuopio Campus, Finland. [2]Department of Environmental and Biological Sciences, University of Eastern Finland, Kuopio Campus, Finland. [3]Department of Technical Physics, University of Eastern Finland, Kuopio Campus, Finland. [4]Institute of Biomedicine, University of Eastern Finland, Kuopio Campus, Finland. [5]Physician and Nursing Services, Health and Social Services Centre, Wellbeing Services County of North Savo, Varkaus, Finland. [6]Faculty of Sport and Health Sciences, University of Jyväskylä, Jyväskylä, Finland. [7]School of Pharmacy, Faculty of Health Sciences, University of Eastern Finland, Kuopio Campus, Finland. [8]LC-MS Metabolomics Center, Biocenter Kuopio, Kuopio, Finland. [9]Food Sciences Unit, Department of Life Technologies, University of Turku, Turku, Finland. [10]Department of Clinical Physiology and Nuclear Medicine, Kuopio University Hospital, Kuopio, Finland. [11]Kuopio Research Institute of Exercise Medicine, Kuopio, Finland. [12]These authors jointly supervised this work: Marjukka Kolehmainen, Timo A. Lakka. ✉e-mail: darren.healy@uef.fi

sum scores in nutrition research include the Mediterranean Diet Score[20] and the Baltic Sea Diet Score[21].

Many of the existing health-related sum scores have focused on physiological risk factors as opposed to external exposures[22,23], with these scores often incorporating physiological variables, such as body mass index (BMI), blood pressure and blood lipids into the score. These physiological variables measure an individual's response to environmental exposures and not the exposures themselves and may therefore represent biological processes through which combined exposures exert their adverse health effects. In recent years, a child healthy lifestyle score was constructed for a Spanish birth cohort, however, it only incorporated specific lifestyle behaviours that were assessed via questionnaires filled out by parents[24]. In this study, we aimed to compare the outcomes of investigating the combined effect of multiple exposures versus individual exposure categories. To do this, we calculated the magnitude of exposure to diet, activity, sleep, air pollution and socioeconomic status. These individual scores were summed together to create a combined environmental and lifestyle exposure ("exposome") score. To explore the exposome score, and to facilitate the comparison with individual exposure scores, we investigated associations of the exposome score with serum metabolites measured using LC-MS and NMR in a general population of children to measure the overall impact of multiple exposures.

The advancement of metabolomics provides a platform to investigate the metabolic effects of different exposures, which can help facilitate the identification of key environmental and lifestyle exposures and elucidate the biological mechanisms underlying their role in cardiometabolic health since childhood[25–27]. Importantly, metabolomics enables the detection of environmental and lifestyle exposures which reflect early metabolic changes prior to the clinical onset of cardiometabolic diseases[25–27]. The application of metabolomics in understanding the exposome has already been proposed[28], while metabolomics has been used in exposure research, with numerous reviews supporting the notion of a metabolic response with exposure to air pollution[29,30], physical activity[31,32], diet[33,34], and sleep[35,36], while it has also been demonstrated that socioeconomic status can affect the human metabolome[37]. This is further supported with findings from a systematic review suggesting that combined healthy lifestyle behaviours are associated with a unique metabolic profile[38]. Two commonly used methodologies in metabolomics are liquid chromatography–mass spectrometry (LC-MS) and nuclear magnetic resonance (NMR), each with their own advantages and disadvantages[39]. LC-MS is highly sensitive and can detect a large range of metabolites in very low levels in biological samples. In contrast, NMR can measure larger compounds e.g., lipoprotein particles that are not amenable for analysis in general LC-MS procedures[39,40]. The combination of both approaches would allow their complementary strengths to be utilised and improve coverage of the metabolome[40,41]. Despite this, very few studies have incorporated both LC-MS and NMR metabolomics data into their analyses[41].

We hypothesise that the net effect of multiple exposures, assessed via the use of a composite exposure ("exposome") score, will capture different associations with serum metabolites measured by LC-MS and NMR than the individual exposure components of the score. The exposome score was associated with 31 metabolites measured by LC-MS and NMR. Of these 31 metabolites, there were 12 metabolites that were not associated with any individual category of the exposome score. These findings support the idea of evaluating the combined effect of multiple exposures when investigating overall health impacts of environmental and lifestyle factors. These findings may also help increase our understanding of the biological effects of combined environmental exposures on metabolic health since childhood.

## Results

### Participant characteristics
Participant characteristics from baseline, 2-year follow-up, and 8-year follow-up are presented in Table 1. Mean sedentary time, parental education, household income, and concentration of the air pollutant nitrogen oxide increased, while mean total physical activity and sleep duration decreased over eight years. Mean concentration of the air pollutant nitrogen dioxide

increased from baseline to 2-year follow-up and decreased back to baseline values at the 8-year follow-up. Despite these changes in exposure variables, the mean exposome score did not change over time.

### Associations of the exposome score with serum metabolites measured by LC-MS
The exposome score was positively associated with seven and negatively associated with nine of all 186 serum metabolites measured by LC-MS over eight years and found to be affected by a lifestyle intervention in children (Table 2). The exposome score was significantly associated with seven glycerophospholipids; of these, the exposome score was positively associated with lysophosphatidylethanolamine 18:2/0:0 (LPE 18:2/0:0, $p = 0.030$), and negatively associated with lysophosphatidylcholine 18:0/0:0 (LPC 18:0/0:0, $p = 0.025$), lysophosphatidylcholine 20:1 (LPC 20:1, $p = 0.030$), lysophosphatidylcholine 22:6 (LPC 22:6, $p = 0.039$), phosphatidylcholine 40:6 (PC 40:6, $p = 0.007$), phosphatidylcholine 34:3 (PC O-34:3)/phosphatidylcholine 34:2 (PC P-34:2) ($p = 0.005$), and phosphatidylethanolamine 16:0/20:4 (PE P-16:0/20:4, $p = 0.020$). The exposome score was negatively associated with linoleamide ($p = 0.011$), a fatty amide. Of the amino acids and their derivatives, the exposome score was positively associated with sarcosine ($p = 0.002$) and indolelactic acid ($p = 0.048$) and negatively associated with 3-aminoisobutanoic acid ($p = 0.046$). The exposome score was positively associated with three xenobiotic compounds: hydroxyferulic acid ($p = 0.015$) and [2 or 6] hydroxybenzothiazole ($p = 0.016$) and an O-glycosyl compound ($C_{15}H_{28}O_{11}$, $p = 0.048$). The exposome score was positively associated with a sterol ($C_{29}H_{46}O_2$, $p = 0.030$). All associations between the exposome score and serum metabolites measured by LC-MS are presented in Supplementary Data 1.

### Associations of the exposome score with serum metabolites measured by NMR
The exposome score was positively associated with 14 and negatively associated with one of all 56 serum metabolites measured by NMR over eight years and found to be affected by a lifestyle intervention in children (Table 3). The exposome score was positively associated with three measures of triglycerides: for every one-unit increase in the exposome score, total triglycerides increased by 0.0098 mmol/L ($\beta = 0.0098$, 95% CI [0.0015, 0.0182], $p = 0.019$), triglycerides in low-density lipoprotein (LDL) by 0.0005 mmol/L ($\beta = 0.0005$, 95% CI [0.00001, 0.0011], $p = 0.044$), and triglycerides in very-low-density lipoprotein (VLDL) by 0.0082 mmol/L ($\beta = 0.0082$, 95% CI [0.0011, 0.0152], $p = 0.021$). Of the measures of fatty acids, the exposome score was positively associated with monounsaturated fatty acids ($\beta = 0.0178$, 95% CI [0.0055, 0.0302], $p = 0.004$), saturated fatty acids ($\beta = 0.0177$, 95% CI [0.0036, 0.0317], $p = 0.013$) and total fatty acids ($\beta = 0.0440$, 95% CI [0.0051, 0.0829], $p = 0.026$), while it was negatively associated with acetate ($\beta = -0.0008$, 95% CI [−0.0015, −0.0002], $p = 0.010$). The exposome score was positively associated with one measure of cholesterol, with increases of 0.0047 mmol/L observed in VLDL-cholesterol for every one-unit increase in the exposome score ($\beta = 0.0047$, 95% CI [0.0004, 0.0089], $p = 0.030$). The exposome score was positively associated with phospholipids in VLDL, with an increase of 0.0031 mmol/L for every one-unit increase in the exposome score ($\beta = 0.0031$, 95% CI [0.0004, 0.0058], $p = 0.023$). The exposome score was positively associated with four amino acids and their derivatives, including alanine ($\beta = 0.0034$, 95% CI [0.0016, 0.0053], $p < 0.001$), creatinine ($\beta = 0.2616$, 95% CI [0.0803, 0.4430], $p = 0.004$), isoleucine ($\beta = 0.0003$, 95% CI [<0.0001, 0.0007], $p = 0.034$) and glycine ($\beta = 0.0016$, 95% CI [0.0001, 0.0030], $p = 0.027$). The exposome score was positively associated with pyruvate, with an increase of 0.0005 mmol/L for every one-unit increase in the exposome score ($\beta = 0.0005$, 95% CI [< 0.0001, 0.0010], $p = 0.035$). The exposome score was positively associated with glycoprotein acetyls, with an increase of 0.0058 mmol/L for every one-unit increase in the exposome score ($\beta = 0.0058$, 95% CI [0.0025, 0.0090], $p < 0.001$). All associations between the exposome score and serum metabolites measured by NMR are presented in Supplementary Data 2.

**Table 1 | Characteristics of participants throughout the PANIC study**

| Characteristic | Baseline (*n* = 504) | 2-year follow-up (*n* = 438) | 8-year follow-up (*n* = 277) | *p*-value |
|---|---|---|---|---|
| Age (years) | 7.63 ± 0.39 | 9.75 ± 0.43 | 15.80 ± 0.43 | <0.001 |
| Sex | | | | 0.649 |
| Female | 243 (48.0%) | 214 (49.0%) | 126 (45.0%) | |
| Male | 261 (52.0%) | 223 (51.0%) | 151 (55.0%) | |
| Pubertal status | | | | <0.001 |
| Tanner Stage 1 | 492 (97.6%) | 322 (77.0%) | 0 (0.0%) | |
| Tanner Stage 2 | 12 (2.4%) | 98 (23.0%) | 0 (0.0%) | |
| Tanner Stage 3 | 0 (0.0%) | 0 (0.0%) | 21 (8.7%) | |
| Tanner Stage 4 | 0 (0.0%) | 0 (0.0%) | 139 (58.0%) | |
| Tanner Stage 5 | 0 (0.0%) | 0 (0.0%) | 81 (34.0%) | |
| Serum cotinine | 2.64 ± 0.43 | 2.59 ± 0.44 | 3.15 ± 1.23 | <0.001 |
| BMI-SDS | −0.17 ± 1.08 | −0.13 ± 1.06 | −0.05 ± 1.02 | 0.297 |
| Finnish Children Healthy Eating Index | 22.41 ± 6.51 | 22.49 ± 7.29 | 22.72 ± 6.37 | 0.867 |
| Total physical activity (min/day) | 626.04 ± 130.90 | 505.13 ± 108.79 | 366.33 ± 131.58 | <0.001 |
| Sedentary time (min/day) | 232.43 ± 129.65 | 382.77 ± 106.32 | 606.76 ± 134.71 | <0.001 |
| Sleep duration (h/day) | 9.65 ± 0.50 | 9.15 ± 0.56 | 7.63 ± 0.76 | <0.001 |
| Parental education | | | | 0.050 |
| Vocational school or less | 98 (20.0%) | 64 (15.0%) | 35 (14.0%) | |
| Vocational high school | 224 (45.0%) | 201 (47.0%) | 100 (40.0%) | |
| University | 179 (36.0%) | 165 (38.0%) | 112 (45.0%) | |
| Household income | | | | <0.001 |
| <€30,000 | 106 (21.0%) | 70 (16.0%) | 22 (9.0%) | |
| €30,000–€60,000 | 206 (42.0%) | 156 (37.0%) | 58 (24.0%) | |
| >€60,000 | 182 (37.0%) | 201 (47.0%) | 162 (67.0%) | |
| Nitrogen oxide (µg/m$^3$) | 6.78 ± 5.80 | 11.12 ± 9.70 | 11.77 ± 6.43 | <0.001 |
| Nitrogen dioxide (µg/m$^3$) | 16.45 ± 7.31 | 24.44 ± 11.75 | 15.06 ± 6.57 | <0.001 |
| Particulate matter ≤ 2.5 µm (µg/m$^3$) | 7.21 ± 3.92 | 6.85 ± 3.21 | 6.55 ± 6.83 | 0.148 |
| Particulate matter ≤ 10 µm (µg/m$^3$) | 14.87 ± 11.26 | 17.57 ± 13.96 | 14.16 ± 12.35 | <0.001 |
| Ozone (µg/m$^3$) | 46.54 ± 17.94 | 43.91 ± 16.64 | 51.03 ± 15.51 | <0.001 |
| Exposome score | 12.51 ± 2.49 | 12.43 ± 2.54 | 11.87 ± 3.05 | 0.094 |

Values are unadjusted means ± standard deviations for continuous variables and numbers (percentages) for categorical variables. *P*-values have been calculated using Kruskal–Wallis rank-sum test or Pearson's Chi-squared test. Serum cotinine values presented here been log$_{10}$ transformed. The exposome score (ranging between 5 and 20) was computed by summing up quartile scores for diet quality (measured by Finnish children healthy eating index), activity (combined score for total physical activity and sedentary time), pollution (combined score for nitrogen oxide, nitrogen dioxide, particulate matter ≤10 µm, particulate matter ≤2.5 µm and ozone), sleep (measured by sleep duration) and socioeconomic status (combined score for household income and parental education).
*BMI-SDS* body mass index standard deviation score.

### Comparison of the associations of the exposome score and its individual scores with serum metabolites measured by LC-MS and NMR

In total, the exposome score or its individual scores were associated with 99 serum metabolites measured by LC-MS (Supplementary Data 3). Of the 16 significant associations between the exposome score and metabolites measured by LC-MS, the exposome score was associated with five metabolites that were not associated with any individual score, including positive associations with [2 or 6] hydroxybenzothiazole (*p* = 0.015), hydroxyferulic acid (*p* = 0.004), an O-glycosyl compound (C$_{15}$H$_{28}$O$_{11}$, *p* = 0.048) and indolelactic acid (*p* = 0.048) and a negative association with LPC 20:1 (*p* = 0.025) (Supplementary Data 3). Individual exposure categories were significantly associated with 84 metabolites that were not significantly associated with the exposome score. Of these, there were 12 metabolites that were significantly associated with two or more of the individual exposure categories, yet not significantly associated with the exposome score (Supplementary Data 3).

In total, the exposome score or its individual scores were associated with 27 serum metabolites measured by NMR (Supplementary Data 4). Of the 15 significant associations between the exposome score and serum metabolites measured by NMR, the exposome score was associated with seven metabolites that were not associated with any individual score, being positively associated with creatinine (*p* = 0.005), isoleucine (*p* = 0.035), monounsaturated fatty acids (*p* = 0.005), saturated fatty acids (*p* = 0.013), total fatty acids (*p* = 0.026), VLDL cholesterol (*p* = 0.030), and VLDL phospholipids (*p* = 0.023) (Supplementary Data 4). Individual exposure categories were significantly associated with 12 metabolites that were not significantly associated with the exposome score (Supplementary Data 4).

### BMI-SDS as a mediator for the associations of the exposome score with serum metabolites measured by LC-MS and NMR

The exposome score was positively associated with BMI-SDS (β = 0.0269, 95% CI [0.0063, 0.0476, *p* = 0.010). The association of the exposome score with serum metabolites measured by LC-MS was weakened after further adjustment for BMI-SDS for eight of the 18 serum metabolites (Supplementary Data 1). The associations of the exposome score with hydroxyferulic acid (*p* = 0.294), linoleamide (*p* = 0.639), PC 40:6 (*p* = 0.063), indolelactic acid (*p* = 0.107), an O-glycosyl compound (C$_{15}$H$_{28}$O$_{11}$, *p* = 0.210) and a sterol (C$_{29}$H$_{46}$O$_2$, *p* = 0.278) were no longer significant after further adjustment for BMI-SDS (Supplementary Data 1). After further

**Table 2 | Associations between the exposome score and serum metabolites measured by LC-MS**

| Annotated metabolite | β (95% CI) | p value |
|---|---|---|
| **Amino acids & derivatives** | | |
| Sarcosine | 3979 (1389, 6569) | 0.002 |
| 3-Aminoisobutanoic acid | −909 (−1806, −12) | 0.016 |
| Indolelactic acid | 2073 (16, 4130) | 0.048 |
| **Fatty amides** | | |
| Linoleamide | −1,154,566 (−2,181,194, −127,939) | 0.027 |
| **Phospholipids** | | |
| Lysophosphatidylcholine 18:0/0:0 | −611,031 (−1,146,923, −75,139) | 0.025 |
| Lysophosphatidylcholine 20:1 | −28,083 (−53,488, −2679) | 0.030 |
| Lysophosphatidylcholine 22:6 | −749 (−1463, −36) | 0.039 |
| Lysophosphatidylethanolamine 18:2/0:0 | 2318 (217, 4419) | 0.030 |
| Phosphatidylcholine 40:6 | −286,071 (−496,681, −75,460) | 0.007 |
| Phosphatidylcholine O-34:3/Phosphatidylcholine P-34:2 | −559,186 (−951,987, −166,384) | 0.005 |
| Phosphatidylethanolamine P-16:0/20:4 | −91,229 (−168,566, −13,893) | 0.020 |
| **Sphingolipids** | | |
| Sphingomyelin d18:1/12:0 | −364 (−704, −23) | 0.036 |
| **Sterols** | | |
| Sterol with formula $C_{29}H_{46}O_2$ | 12,241 (1122, 23,360) | 0.030 |
| **Xenobiotic** | | |
| [2 or 6] Hydroxybenzothiazole | 410,580 (76,645, 744,515) | 0.016 |
| O-glycosyl compound with formula $C_{15}H_{28}O_{11}$ | 40,577 (451, 80,703) | 0.047 |
| Hydroxyferulic acid | 7215 (1353, 13,078) | 0.015 |

Unstandardized regression coefficients β (95% confidence intervals) are for estimated changes in metabolite abundances for each one-unit increase in the exposome score over eight years, and p-values are from linear mixed-effects models adjusted for time, sex, age, pubertal stage, and serum cotinine.

*LC-MS* liquid chromatography–mass spectrometry.

adjustment for BMI-SDS, the exposome score became significantly associated with 85 serum metabolites that were not previously associated with the exposome score (Supplementary Data 1).

The association of the exposome score with serum metabolites measured by NMR was weakened after further adjustment for BMI-SDS for 14 of the 15 serum metabolites (Supplementary Data 2). The associations of the exposome score with serum metabolites measured by NMR were partly explained by BMI-SDS, including for monounsaturated fatty acids (by 13.8%), saturated fatty acids (10.5%), total fatty acids (15.6%), LDL triglycerides (9.3%), VLDL triglycerides (24.7%), total triglycerides (21.7%), VLDL cholesterol (25.4%), VLDL phospholipids (26.1%), alanine (8.8%), creatinine (7.8%), isoleucine (9.8%) and glycoprotein acetyls (24.9%), acetate (4.7%) and pyruvate (2.0%) (Supplementary Data 2).

**BMI-SDS as a modifier for the associations of the exposome score with serum metabolites measured by LC-MS and NMR**

The association of the exposome score was significantly modified by BMI-SDS for 12 of the 186 serum metabolites measured by LC-MS and found to be affected by a lifestyle intervention in children (Supplementary Data 5). Six metabolites were found to be increased with a higher exposome score in individuals with higher BMI-SDS compared to those with lower BMI-SDS; this occurred either by an increase in individuals with higher BMI-SDS, a decrease in individuals with lower BMI-SDS, or a combination of both. BMI-SDS modified the associations of the exposome score with 5-aminovaleric acid betaine ($p = 0.027$ for the interaction between BMI-SDS and the exposome score), creatinine ($p = 0.045$), hexanoylcarnitine ($p = 0.018$), hydroxyferulic acid ($p = 0.025$), serinyl-alanine peptide ($p = 0.019$) and xanthine ($p < 0.001$) (Supplementary Data 5). In contrast, six metabolites were observed to be decreased with higher exposome score in individuals with higher BMI-SDS compared to those with lower BMI-SDS; this occurred either by a decrease in individuals with higher BMI-SDS, an increase in individuals with lower BMI-SDS, or a combination of both. BMI-SDS modified the associations of the exposome score with LPE 18:2/0:0 ($p = 0.015$), phosphatidylcholine 16:0/20:4 (PC 16:0/20:4, $p = 0.037$), phosphatidylcholine 36:4 (PC 36:4, $p = 0.047$), lysophosphatidylcholine 0:0/18:2 (LPC 0:0/18:2, $p = 0.046$), LPC 18:2/0:0 ($p = 0.022$) and an amino acid derivative (C4H7N5O2, $p = 0.036$) (Supplementary Data 5).

The association of the exposome score was significantly modified by BMI-SDS for 19 of the 56 serum metabolites measured by NMR and found to be affected by a lifestyle intervention in children (Supplementary Data 6). Altogether, 18 metabolites measured by NMR were observed to be increased with a higher exposome score in individuals with higher BMI-SDS compared to those with lower BMI-SDS; this occurred either by an increase in individuals with higher BMI-SDS, a decrease in individuals with lower BMI-SDS, or a combination of both. BMI-SDS positively modified the associations of the exposome score with creatinine ($p = 0.008$), cholines ($p = 0.046$), leucine ($p = 0.016$), valine ($p = 0.004$), total branched chain amino acids ($p = 0.008$), monounsaturated fatty acids ($p = 0.001$), omega-3 fatty acids ($p = 0.039$), omega-6 fatty acids ($p = 0.022$), phosphoglycerides ($p = 0.047$), remnant cholesterol ($p = 0.038$), polyunsaturated fatty acids ($p = 0.016$), linoleic acid ($p = 0.022$), saturated fatty acids ($p = 0.001$), total fatty acids ($p = 0.001$), total triglycerides ($p = 0.006$), total phospholipids ($p = 0.033$), glycerol ($p = 0.028$), and glycoprotein acetyls ($p = 0.002$) (Supplementary Data 6). One metabolite measured by NMR was found to be decreased with higher exposome score in individuals with higher BMI-SDS compared to those with lower BMI-SDS. BMI-SDS negatively modified the association of the exposome score with acetate ($p = 0.023$) (Supplementary Data 6). Selected serum metabolites measured by LC-MS and NMR are presented in Fig. 1 to display the different patterns of the modification of the associations of the exposome score with serum metabolites measured by BMI-SDS.

**Sensitivity analysis**

**Comparison of diet scores within exposome score.** The use of the Finnish Children Healthy Eating Index, Mediterranean Diet Score, or Baltic Sea Diet Score in the formulation of the exposome score was significantly associated with 16, 27 and 20, respectively, serum metabolites measured by LC-MS (Supplementary Data 1). Altogether, 11 metabolites were common to all three models. However, there were four, eight and two unique associations when the Finnish Children Healthy Eating Index, the Mediterranean Diet Score, and the Baltic Sea Diet Score were used in the exposome score, respectively (Supplementary Fig. S1).

The use of the Finnish Children Healthy Eating Index, the Mediterranean Diet Score, or the Baltic Sea Diet Score in the formulation of the exposome score was significantly associated with 15, 11 and 22, respectively, serum metabolites measured by NMR (Supplementary Data 2). Four metabolites were common to all three models. However, there were five, two and seven unique associations when the Finnish Children Healthy Eating Index, the Mediterranean Diet Score, and the Baltic Sea Diet Score were used in the exposome score, respectively (Supplementary Fig. S2).

**Leave-one-out analysis.** The original exposome score model and leave-out models were associated with 54 serum metabolites measured by LC-MS (Supplementary Data 7). The directions of the significant associations of the exposome score with serum metabolites measured by LC-MS were consistent across the various leave-out models (Supplementary Data 7).

**Table 3 | Associations between the exposome score and serum metabolites measured by NMR**

| Metabolite name | Mean ± SD | β (95% CI) | *p* value |
|---|---|---|---|
| Amino acids & derivatives | | | |
| Alanine | 0.31 ± 0.07 | 0.0034 (0.0016, 0.0053) | <0.001 |
| Creatinine | 47.74 ± 12.89 | 0.2616 (0.0803, 0.4430) | 0.004 |
| Glycine | 0.28 ± 0.05 | 0.0016 (0.0001, 0.0030) | 0.027 |
| Isoleucine | 0.05 ± 0.03 | 0.0003 (<0.0001, 0.0007) | 0.034 |
| Cholesterol | | | |
| VLDL-cholesterol | 0.47 ± 0.15 | 0.0047 (0.0004, 0.0089) | 0.030 |
| Fatty acids | | | |
| Acetate | 0.05 ± 0.02 | −0.0008 (−0.0015, −0.0002) | 0.010 |
| Monounsaturated fatty acids | 2.42 ± 0.44 | 0.0178 (0.0055, 0.0302) | 0.004 |
| Saturated fatty acids | 3.5 ± 0.51 | 0.0177 (0.0036, 0.0317) | 0.013 |
| Total fatty acids | 10.51 ± 1.41 | 0.0440 (0.0051, 0.0829) | 0.026 |
| Glycolysis related metabolites | | | |
| Pyruvate | 0.05 ± 0.02 | 0.0005 (<0.0001, 0.0010) | 0.035 |
| Inflammatory protein markers | | | |
| Glycoprotein acetyls | 0.76 ± 0.1 | 0.0058 (0.0025, 0.0090) | <0.001 |
| Phospholipids | | | |
| VLDL-phospholipids | 0.27 ± 0.09 | 0.0031 (0.0004, 0.0058) | 0.023 |
| Triglycerides | | | |
| LDL-triglycerides | 0.11 ± 0.02 | 0.0005 (<0.0001, 0.0011) | 0.044 |
| Total triglycerides | 0.74 ± 0.28 | 0.0098 (0.0015, 0.0182) | 0.019 |
| VLDL-triglycerides | 0.47 ± 0.24 | 0.0082 (0.0011, 0.0152) | 0.021 |

Unstandardized regression coefficients β (95% confidence intervals) are for estimated changes in metabolite abundances for each one-unit increase in the exposome score over eight years, and *p* values are from linear mixed-effects models adjusted for time, sex, age, pubertal stage, and serum cotinine.
*NMR* nuclear magnetic resonance, *LDL* low-density lipoprotein, *VLDL* very-low-density lipoprotein.

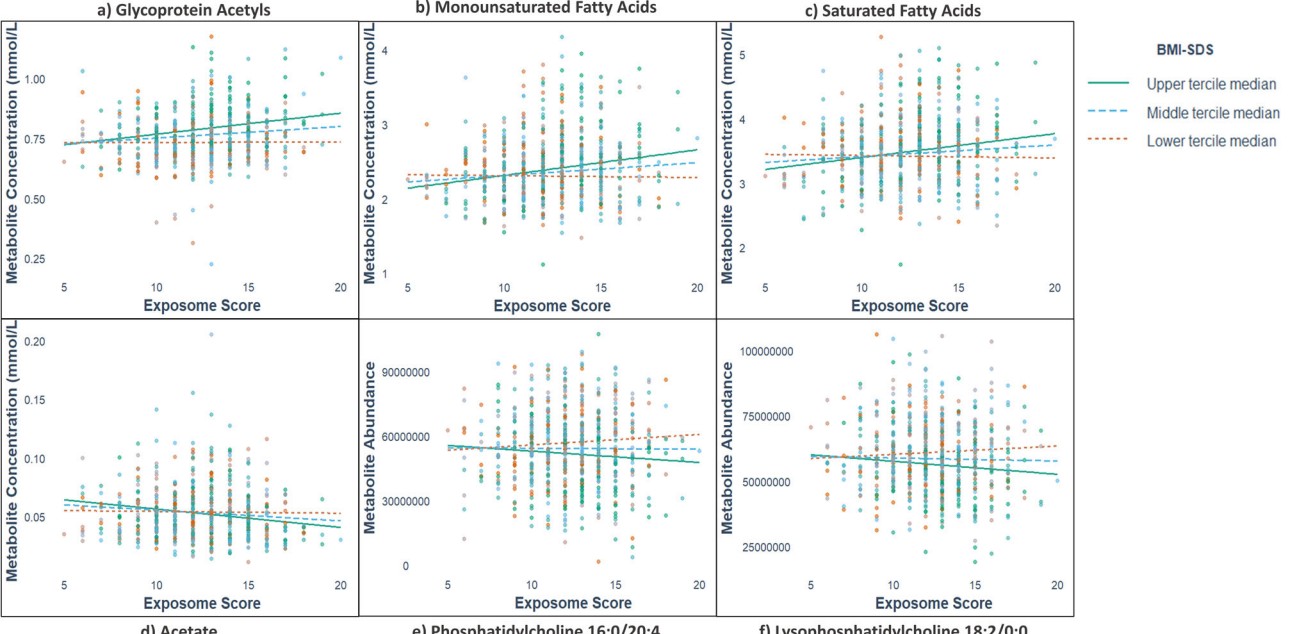

**Fig. 1 | Modification of the association of the exposome score with selected serum metabolites measured by LC-MS and NMR by BMI-SDS.** This figure illustrates the different patterns of how adiposity, measured by BMI-SDS, modified the associations of the exposome score with serum metabolites measured by NMR (**a–d**) and LC-MS (**e**, **f**). Panels **a**, **b** and **c** show metabolite concentration increased in individuals with higher BMI-SDS, while panels **d**, **e** and **f** show metabolite abundance/concentration decreased in individuals with higher BMI-SDS. The exposome score (ranging between 5 and 20) was computed by summing up quartile scores for diet quality (measured by Finnish children healthy eating index), activity (combined score for total physical activity and sedentary time), pollution (combined score for nitrogen oxide, nitrogen dioxide, particulate matter ≤10 μm, particulate matter ≤2.5 μm and ozone), sleep (measured by sleep duration) and socioeconomic status (combined score for household income and parental education). BMI-SDS body mass index standard deviation score, LC-MS liquid chromatography–mass spectrometry, NMR nuclear magnetic resonance. Medians for BMI-SDS terciles: Lower = −1.2, Middle = −0.1 and Upper = 0.8.

Most of the models were observed to have a similar number of significant associations with metabolites as the original exposome score model, ranging between 13 and 16 (Supplementary Fig. S3). The deviation to this trend was found with the removal of the air pollution category from the exposome score; this model had 29 significant associations with metabolites, which was the largest number of such associations (Supplementary Fig. S3, Supplementary Data 7).

The original exposome score model and leave-out models were associated with 27 serum metabolites measured by NMR (Supplementary Data 8). The directions of the significant associations of the exposome score with serum metabolites measured by NMR were consistent across the various leave-out models (Supplementary Data 8). In comparison with the original exposome score model, the leave-out models for diet, activity, sleep, and socioeconomic status were all significantly associated with less metabolites (8, 5, 12 and 8, respectively). The exception to this was the leave-out-model for pollution, which was significantly associated with 23 metabolites (Supplementary Fig. S4, Supplementary Data 8).

## Discussion

Here, we showed that the combination of environmental and lifestyle exposures, including diet quality, activity, air pollution, sleep duration, and socioeconomic status, was associated with numerous serum metabolites measured by LC-MS and NMR in childhood and adolescence. Furthermore, we demonstrated that this exposome score was uniquely associated with 12 serum metabolites, corresponding to 39% of the total number of associations between the exposome score and serum metabolites. We also found that adiposity mediated and modified many of the associations of the exposome score with serum metabolites. To the best of our knowledge, this is the first longitudinal study to investigate the association between an exposome score and metabolic health across childhood and adolescence.

The exposome score was associated with 16 serum metabolites measured by LC-MS in childhood and adolescence, with many of these metabolites observed to be decreased in individuals with a higher exposome score. We also see that, as the exposome score increases, the concentration of most serum metabolites measured by NMR are increased. Across both LC-MS and NMR metabolomics data, the exposome score was associated predominantly with serum lipids, amino acids, and their derivatives. Altered lipid metabolism has been associated with increased cardiometabolic risk in adults[42] and in children living with obesity[43], with increased circulating glycerophosphocholines and triglycerides implicated in both. In our study, all the associations of the exposome score with serum lipids measured by LC-MS were glycerophospholipids, with all noted to be decreased in individuals with a higher exposome score. Glycerophospholipids are a group of lipids that have been shown to have neuroprotective and anti-inflammatory effects[44–46]. In addition to these findings, acetate, a short-chain fatty acid produced by the gut microbiota, was the only metabolite measured by NMR with which the exposome score was negatively associated. There is a large body of mechanistic evidence from animal studies that suggest acetate can improve cardiometabolic health through antilipolytic and anti-inflammatory effects[47], while acetate supplementation has been shown to prevent the development of hypertension and reduce neuroinflammation in mice[48,49]. Interestingly, the negative associations of the exposome score with several compounds that exhibit anti-inflammatory effects coincide with a positive association between the exposome score and serum glycoprotein acetyls measured by NMR. Glycoprotein acetyls are a biomarker of systemic inflammation and metabolic dysfunction[50–52], and are suggested to provide a sensitive measure to detect increased cardiovascular risk in young people[53] and predict cardiometabolic risk in adulthood[54]. Along with triglycerides, total fatty acids and monounsaturated fatty acids, glycoprotein acetyls have also been directly associated with the risk of type 2 diabetes in Asian and European adults[55].

Inflammation can lead to altered lipid metabolism[56], with inflammation shown to modify circulating levels of lipoproteins and triglycerides in children and adolescents[57]. In this study, along with the positive association of the exposome score with serum glycoprotein acetyls, a measure of systemic inflammation, we found that the exposome score was positively associated with various serum lipid metabolites measured by NMR. These include total, LDL and VLDL triglycerides and total, saturated, and monounsaturated fatty acids. In individuals with a higher exposome score, we also observed elevated serum concentrations of VLDL phospholipids and VLDL cholesterol. Serum concentration of triglycerides[58] and LDL triglycerides[59] have been positively associated with insulin resistance in children, while the Bogalusa Heart Study found that serum concentrations of total triglycerides and LDL cholesterol, among others, had strong positive associations with the development of atherosclerosis in young people[60]. In addition, VLDL cholesterol has been associated with increased risk of coronary heart disease in adults[61,62].

The exposome score was also positively associated with serum alanine, glycine, and isoleucine, measured by NMR. These amino acids measured by NMR have been identified as potential biomarkers of cardiometabolic disorders, including obesity[25] and type 2 diabetes[25], and are involved in various metabolic pathways, such as glycine, serine, and threonine metabolism and valine, leucine and isoleucine metabolism[25]. We also found an increase in sarcosine, as measured by LC-MS, in individuals with a higher exposome score. In a recent study, elevated sarcosine was determined to be a marker of dyslipidemia, with the increase in sarcosine even more pronounced across profiles that included further risk factors for metabolic syndrome[63]. We also found that the exposome score was positively associated with serum pyruvate, which was measured by NMR and is involved in several metabolic processes, such as alanine, glycine, serine, and threonine metabolism, which have been implicated in cardiometabolic diseases[25]. Metabolites of the glycolysis and gluconeogenesis pathways, including pyruvate, have also been implicated in the pathogenesis of obesity and type 2 diabetes in adults[64–66].

Of the 31 significant associations of the exposome score with serum metabolites measured by LC-MS and NMR, the exposome score was associated with 12 serum metabolites that were not associated with any individual score (five measured by LC-MS, seven measured by NMR), all of which were lipids, xenobiotics, and amino acids (Supplementary Data 3–4). As a demonstration for the potential value of using a composite exposure score, we can consider the positive association of the exposome score with [2 or 6] hydroxybenzothiazole measured by LC-MS. In vitro studies have demonstrated that environmentally relevant circulating concentrations of these compounds could lead to increased levels of reactive oxygen species[67,68]. Increased oxidative stress, along with systemic inflammation, is considered one of the primary initiating mechanisms in the development of cardiometabolic diseases[69–71]. Furthermore, benzothiazoles have been shown to interact with aryl hydrocarbon receptors[72], which have been implicated in insulin resistance and impaired glucose homeostasis[73]. [2 or 6] hydroxybenzothiazole is not a naturally occurring metabolite, with benzothiazoles widely used in the manufacture of household and consumer products[74]. Benzothiazoles have also been detected in indoor[75] and outdoor air[76,77], clothing textiles[78], and food and beverages, including seafood[79], tea[80], drinking water[81], and commercial milk products[82]. Despite the fact that benzothiazoles can be found in numerous sources, [2 or 6] hydroxybenzothiazole was not associated with any individual exposure component of the exposome score, while it was found to be positively associated with the exposome score. This underlines the added value of the composite exposome score in depicting elevated serum levels of this potentially harmful metabolite.

The positive associations of the exposome score with monounsaturated, saturated, and total fatty acids measured by NMR can be used as another example of the potential value of composite exposure scores. In adults, higher levels of circulating saturated fatty acids have been associated with an increased risk of cardiometabolic diseases in a meta-analysis by Li

et al.[83], whereas monounsaturated fatty acids were positively associated with risk of coronary artery disease in a large prospective cohort study[84]. The literature on exposure research and fatty acid profiles in children is limited, however, some studies have noted associations of environmental exposures with fatty acid composition and metabolism; exposure to air pollution has been associated with altered fatty acid metabolism in children[85,86], while it has also been suggested that dysregulated lipid and fatty acid metabolism may mediate the associations of per- and polyfluoroalkyl substances with glucose intolerance in young adults[87]. Furthermore, a combined diet and physical activity intervention was reported previously to have a beneficial effect on fatty acid composition in children who participated in the PANIC study[88]. Similarly to [2 or 6] hydroxybenzothiazole, while it appears that levels of fatty acids can be affected by environmental and lifestyle factors, we did not observe associations of any individual exposure component of the exposome score with fatty acids measured by NMR, while they were found to be positively associated with the exposome score. The findings for [2 or 6] hydroxybenzothiazole and fatty acids, along with the other eight serum metabolites that were exclusively associated with the exposome score, highlight the value of using a composite score of multiple environmental and lifestyle exposures when identifying serum biomarkers of potential relevance for cardiometabolic health, given the complex interplay between environmental and lifestyle exposures and biological responses.

Adiposity was found to mediate several of the significant associations of the exposome score with serum metabolites measured by NMR, explaining from 2% to 26% of these associations. Given that energy intake was not incorporated into the diet quality score, it is not surprising that adiposity did not mediate a larger proportion of the associations of the exposome score with serum metabolites. While adiposity also appeared to mediate the association of the exposome score with some metabolites measured by LC-MS, a potentially more relevant finding was the large number of serum metabolites measured by LC-MS (roughly 45%), most of which were phospholipids, that became significantly associated with the exposome score after controlling for adiposity. This contrasted with the associations of the exposome score with serum metabolites measured by NMR, where no new significant associations were observed after controlling for adiposity. It may be that adiposity affects many of the compounds measured by LC-MS through different mechanisms independently of the exposures measured in this study, and future studies should carefully consider its role in the associations of environmental and lifestyle exposures with serum metabolites.

Adiposity was found to modify the associations of the exposome score with a number of serum metabolites in our study. Serum creatinine measured by LC-MS and NMR was observed to be higher in individuals with higher adiposity. While serum creatinine is a commonly used clinical measure of kidney function and marker of increased cardiovascular risk, this is reported in adults[89]. In our general population of children and adolescents, however, none had kidney disease. Higher serum creatinine in our participants is likely to be a measure of skeletal muscle mass because creatinine is mainly produced by skeletal muscle[90]. It has been reported that children with overweight/obesity have higher amounts of skeletal muscle mass compared to children with normal weight[91,92]. The higher levels of creatinine in individuals with higher adiposity may therefore be explained by a combination of increased skeletal muscle mass and the clustering of adverse environmental exposures. We also observed that adiposity modified the association of the exposome score with serum amino acids and their derivatives measured by LC-MS and NMR. Previous research has shown strong associations of BMI with some circulating amino acids such as branched chain amino acids, including leucine, isoleucine and valine[93]. Circulating concentrations of branched-chain amino acids have also been directly associated with future risk of insulin resistance in children and adolescents, and with cardiometabolic risk factors in young adults[94,95]. Further research is required to determine if certain amino acids could be used as biomarkers of disturbed metabolic processes, and whether this only applies to children and adolescents living with overweight. The association between the exposome score and serum glycoprotein acetyls, which has previously been discussed as a biomarker for systemic inflammation and metabolic dysfunction, was

also modified by adiposity, with increased serum glycoprotein acetyls found in individuals with higher adiposity. Previous studies have shown that adiposity is positively associated with circulating biomarkers of inflammation in children[96,97]. Furthermore, higher levels of sedentary time and lower levels of physical activity were associated with higher circulating levels of inflammatory biomarkers in children with increased adiposity, which appears concordant with our results[98]. This suggests that individuals with higher adiposity are more susceptible to the inflammatory effects of environmental and lifestyle exposures deemed to be unhealthy. In individuals with a low exposome score, regardless of adiposity, concentration of serum glycoprotein acetyls did not appear to differ. It may be the case that individuals living with overweight may not suffer increased inflammation should they mitigate their unhealthy exposures, such as adopting a healthier lifestyle comprised of improved diet quality and increased physical activity; it has previously been shown that physical fitness is negatively associated with concentration of circulating inflammatory biomarkers independently of BMI[99].

Measuring the exposome is a major challenge, and there have been efforts in recent years to measure multiple exposures and try to account for the interactions between them more accurately[100]. One approach that has been implemented in other fields is the use of sum scores, such as in dietary research[20,21] and the development of various health scores[17,18]. While composite scores are widely used in research[101,102], there are limitations to their use in clinical research[102], many of which are also relevant to exposome research. Despite the limitations, it is argued that the use of scores allows the net effect of the overall factors to be assessed[103,104]. To demonstrate the challenges of developing a composite exposure score and exploring its feasibility, we conducted a sensitivity analysis using just one component of the exposome score (diet), where the use of different, validated, approaches to assess diet quality were compared. The original exposome score model, using the Finnish Children Healthy Eating Index, was compared to two sensitivity exposome models using either the Baltic Sea Diet Score or the Mediterranean Diet Score. While there was some overlap between the three models, there were also a number of unique associations with serum metabolites measured by LC-MS and NMR based on the diet score incorporate into the exposome score (Supplementary Figs. S1–S2). Interestingly, when we compared the results of the sensitivity analyses using the Finnish Children Healthy Eating Index and the Baltic Sea Diet Score, both of which have been validated in a Finnish population[20,21], we found that almost a third of the metabolites measured by LC-MS and NMR that were associated with the original exposome score were not found to be related to the exposome score with the Baltic Sea Diet Score. Future research should consider the suitability of selected measures to the population studied, and further research is necessary to determine optimal approaches to assess the impact of diet on an individuals' overall exposome, whether this be through a diet quality approach, or the assessment of specific foods or nutrients. Future research should also explore the impact of ultra-processed foods, which have been positively associated with obesity and other cardiometabolic risk factors in children and adolescents[105]. Specifically, animal-based products and sugar-sweetened beverages should be examined, which were recently determined as the main subgroups of ultra-processed foods associated with cardiometabolic multimorbidity in adults across 10 European countries[106]. These are insights based on the diet component of the exposome score, with special care likely needed for all components incorporated into a composite exposure score. This highlights the need to carefully reflect upon and select appropriate measures, and thus researchers should be careful in the planning of composite scores to measure multiple exposures, and the approaches used to assess the components to be incorporated.

Another limitation of the use of composite scores is that it risks oversimplifying the relationships between the various components incorporated into the score. To assess the contribution of the individual exposure components to the exposome score, we conducted a leave-one-out sensitivity analysis. No individual component appeared to be influencing the directions of the associations, with the direction remaining consistent between the exposome score and the various leave-out models. However, it appeared

that the air pollution score had a large influence on the number of significant associations, with the leave-out model for pollution significantly associated with the largest number of metabolites measured by both LC-MS and NMR. Interestingly, when we inspect metabolites that were significantly associated with the pollution score and at least one other exposure component, we can see that the directions differ in most instances. It is possible that the pollution score is exerting a small effect in the opposing direction to the other exposures, and while not a large enough effect to be significant, it may be enough to influence the overall exposome score, as evidence by the comparison of the original exposome score with the exposome score from which the air pollution component was removed. The assumption that there is a consistent direction for the various factors incorporated into scores, known as the assumption of uniform directionality, is a disadvantage of a scoring approach[107]. This remains an issue for many of the newer methods that try to measure the mixture effect of various exposures[100], although there are methods being developed that try to accommodate the opposing directions of different exposures[108].

Our study has several strengths. The general population of children followed up for eight years, the use of linear mixed-effects models with three time points, and the availability of several environmental and lifestyle factors and numerous metabolites measured by LC-MS and NMR allowed us to analyse the longitudinal associations of the exposome score with serum metabolome from childhood to adolescence. Furthermore, we used objective measures for the variables included in the exposome score—for example, pollutant concentrations were actual measurements from monitoring stations in Kuopio, while physical activity, sedentary time and sleep were all measured using wearable monitors. Furthermore, measuring serum metabolites via both LC-MS and NMR provided us with different insights and a better overall understanding of the overall biological impacts of the environmental and lifestyle exposures. However, there were also a few limitations in our study. Firstly, while the composite exposome score allows for multiple external exposures to be combined and assessed, we did not incorporate any weighting, and thus each environmental and lifestyle score was considered to contribute equally to the overall effect; this was an issue both within-score and between-score. Furthermore, the scoring approach assumed a linear relationship for all variables. It is not always the case that "more is worse/better"—for example, with sleep, both too little and too much have been associated with impaired health. Another limitation with the composite exposure score was that the only external environmental exposures included were all air pollutants, and future research should consider the impact of other factors such as temperature, green and blue spaces, urbanization, and noise. Furthermore, since the objective was to explore the feasibility and value of a composite exposure score, the analyses were carried out with a subset of metabolites already identified as being changed by the intervention. Using a narrow subset of metabolites leaves out other, potentially relevant metabolites that are not affected by the intervention but important in the wider exposome context. Due to issues with data missingness, it was not possible to explore the role of early-life exposures, and subsequent studies should consider early-life exposures, such as maternal exposures during pregnancy, in the context of the exposome. It should also be noted that there was a large number of dropouts, around 45% of the number of participants at baseline, which may introduce bias to the study. Finally, when assessing the residuals of the linear mixed-effects models for several LC-MS metabolites, it appeared that there may be a predictor missing that would help explain some of the variance, which was not adjusted for in the models.

In conclusion, the exposome score, comprised of environmental and lifestyle exposures, including diet quality, activity, sleep duration, air pollution and socioeconomic status, was associated with a large number of serum metabolites measured by LC-MS and NMR in childhood and adolescence. These metabolites were predominantly phospholipids, fatty acids, amino acids, xenobiotics, and energy-related metabolites. Furthermore, the exposome score was uniquely associated with several of these metabolites, highlighting the value of composite scores in predicting metabolic changes associated with multiple environmental and lifestyle exposures since

childhood. However, further research is required on the feasibility of a general exposome score that can be used across populations. We have also investigated associations with individual metabolites, and it may be worthwhile to explore metabolomic profiles attributed to different exposome patterns. Furthermore, due to differing exposure patterns as well as the different impacts the various exposures may have on children compared to adults, the use of composite exposure scores in adults still needs to be explored.

## Methods

### Study design and participants

The Physical Activity and Nutrition in Children (PANIC) study aims to investigate the effects of an individualised and family-based physical activity and dietary intervention on cardiometabolic risk factors and other health outcomes in a general population of children aged 6–9 years followed up for eight years until adolescence. The Research Ethics Committee of the Hospital District of Northern Savo approved the study protocol in 2006 (Statement 69/2006). The PANIC study is registered at ClinicalTrials.gov with the identifier NCT01803776. At baseline and 2-year follow-up, the caregivers gave their written informed consent, and the children provided their assent to participation. At 8-year follow-up, both caregivers and adolescents gave their written informed consent. The PANIC study has been carried out in accordance with the principles of the Declaration of Helsinki as revised in 2008. All ethical regulations relevant to human research participants were followed.

Study design, recruitment, and participants have been described in detail elsewhere[109,110]. In short, 504 children, aged 6–9 years, from the city of Kuopio, Finland, participated in baseline examinations between October 2007 and December 2009. The participants were examined again two and eight years later. A total of 438 children participated in the 2-year follow-up examinations between 2009 and 2011, and 277 adolescents attended the 8-year follow-up examinations between 2015 and 2017.

### Metabolomics analyses

Non-targeted metabolite profiling of fasting serum samples collected at baseline, 2-year follow-up, and 8-year follow-up examinations was performed using LC-MS as described previously[111]. Samples were analysed utilising four LC-MS methods. Reversed-phase analyses were performed using a ultra-high performance liquid chromatography (UHPLC)-Orbitrap-mass spectrometry system (Thermo Fischer Scientific, Bremen, Germany), which consisted of a Vanquish UHPLC system using Zorbax Eclipse XDB-C18 column (particle size 1.8 μm, 2.1 × 100 mm; Agilent Technologies), and a Q Exactive Focus mass spectrometer. The hydrophilic interaction liquid chromatography was performed with a UPLC-quadrupole time-of-flight (QTOF)-mass spectrometry system (Agilent Technologies, Santa Clara, CA, USA), which consisted of a 1290 UPLC using Acquity UPLC BEH amide column (2.1 × 100 mm, 1.7 μm; Waters Corp., Milford, MA, USA), and a 6540 UHD QTOF mass spectrometer. Both positive and negative electrospray ionisation were employed, yielding four data files for each sample[111,112].

After obtaining the molecular features using MS-DIAL v4.90 software, where peak picking and alignment are carried out[113], the data were preprocessed using R software (R Core Team, 2021, https://www.R-project.org) and the *notame* package[111]. Data from each of the four analytical modes were processed separately, which included correcting the molecular features for the drift pattern, checking feature quality based on the quality control samples, and imputing the missing values with random forest imputation[111,112]. Once the preprocessing had been completed, the data was used in statistical analyses, to focus on the molecular features observed to be significantly affected by the physical activity and dietary intervention from baseline to 8-year follow-up examinations in the PANIC study, with these compounds selected for identification[112].

Identification of the compounds followed the 4-level annotation scheme proposed by the Metabolomics Standards Initiative[114], with annotation of the molecular features based on ms/ms spectral comparison on databases including METLIN, MassBank of North America, Human

Metabolome Database and LIPID MAPS, as well as published literature. Level 1 identification was based on retention time and ms/ms fragmentation match against in-house purified standards. The metabolites addressed in the present analyses are thus those that were found to be affected by the physical activity and dietary intervention from baseline to 8-year follow-up examinations in the PANIC study[112].

Serum cotinine, a nicotine metabolite measured by LC-MS, was used as a biomarker of environmental exposure to smoking. In addition to the serum metabolites analysed by the LC-MS method, the Nightingale high-throughput NMR platform was used to measure serum metabolites, including those related to lipoprotein, triglyceride, apolipoprotein, fatty acid and amino acid metabolism as well as systemic inflammation[115].

### Assessment of lifestyle factors
Dietary factors were assessed using 4-day food records, with the Finnish Children Healthy Eating Index used to reflect overall diet quality, ranging from 2–45, with a higher score indicative of better overall diet quality[116]. Details of the methods, along with the computation of the index, are explained elsewhere[110]. The Mediterranean Diet Score and the Baltic Sea Diet Score were also computed for the purpose of a sensitivity analysis[20,21], with descriptions of the calculations of the scores described elsewhere[117].

Total physical activity (sum of light, moderate, and vigorous physical activity, measured in minutes per day), sedentary time (defined as time spent at intensity ≤1.5 metabolic equivalents excluding sleep) and sleep duration (hours per night) were measured using the Actiheart heart rate and body movement monitor (CamNTech Ltd, Cambridgeshire, United Kingdom), with the methods explained in detail previously[110,118].

### Assessment of environmental factors
Air pollution was monitored across multiple measurement sites by the city of Kuopio, with the concentrations presented in this study as spatial averages of all sites. Data used for the air pollution variables, including nitrogen oxide, nitrogen dioxide, ozone, particulate matter ≤2.5 μm and particulate matter ≤10 μm, and details of the measurements are accessible via the open data repository of the Finnish Meteorological Institute (https://en.ilmatieteenlaitos.fi/open-data). The averaging was deemed necessary as it was not possible to assign specific concentrations for all participants based on their home address and daily activities. The spatial averaging was possible as the Kuopio area is rather homogeneous in terms of air pollution. Parental education was based on the highest completed or ongoing degree (vocational school or less, vocational high school, university) according to the parent who attained the highest level. Household income was reported by the higher earning caregiver, with participants classified according to gross income (≤€30,000, €30,001–€60,000, >€60,000).

### Assessment of body size and composition
Body height and weight were measured in the morning in a fasted state; height was measured with a calibrated wall-mounted stadiometer to an accuracy of 0.1 cm, while weight was measured using an Inbody 720 bioelectrical impedance device (Biospace, Seoul, South Korea) with an integrated weight scale to an accuracy of 0.1 kg. BMI was calculated using the formula weight (kg)/[height (m)]$^2$. Age- and sex-standardised BMI standard deviation scores (BMI-SDS) were calculated based on Finnish references[119]. Pubertal status was assessed by a research physician according to breast development for girls (scored M1-5) and testicular volume measured by an orchidometer for boys (scored G1-5), and was based on the stages described by Tanner[120,121].

### Computation of exposome score
To explore the overall associations of multiple exposures across childhood and adolescence, an exposome score was created, taking inspiration from approaches such as an "unhealthy lifestyle score" developed by Bekaert et al.[122] and a "low-risk lifestyle score" by Li et al.[19]. Five separate exposure scores were created to compose the exposome score based on available information; diet quality, activity, sleep duration, air pollution, and socioeconomic status. This exposome score ranged between 5 and 20, with a higher score indicating increased exposure to combined unhealthy environmental and lifestyle factors. The diet score was based on the Finnish Children Healthy Eating Index, with a higher index value corresponding to a lower diet score. The activity score incorporated total physical activity and sedentary time; higher total physical activity contributed to a lower activity score, while higher sedentary time contributed to a higher activity score. The sleep score used sleep duration, where lower sleep duration corresponded to a higher (worse) sleep score. Air concentrations of nitrogen oxide, nitrogen dioxide, particulate matter ≤2.5 μm, particulate matter ≤10 μm and ozone were assessed for the air pollution score, with higher pollutant concentrations attributed a higher score. Finally, the socioeconomic status score comprised of parental education and household income, with lower education and income given a higher score.

To construct the exposome score, quartiles were calculated for each category, with a score of one representing the lowest-risk exposure, while a score of four representing the highest-risk exposure. If an exposure category consisted of more than one measurement, for example the pollution category with five air pollution variables, quartile scores were calculated for each measurement. These quartile scores were then summed, and the average score calculated for that category. This differed for the socioeconomic score, as these variables consisted of three categories. In this instance, values of 1–3 were assigned to each category, with 1 corresponding to highest income or education and 3 corresponding to lowest income or education. These values were summed together, with a summed value of 6 corresponding to a socioeconomic score of 1, while summed values of 5, 4, 3 and 2 corresponded to a socioeconomic score of 2, 3, 4 and 4, respectively. Finally, the scores from each component were summed together to generate the exposome score and reflect overall personal exposure.

### Statistical analyses
All statistical analyses were performed using the R software (R Core Team, 2021, https://www.R-project.org) and RStudio, an Integrated Development Environment for R, version 2022.7.1.554 (RStudio, Boston, MA, USA). Descriptive statistics are presented as unadjusted means ± standard deviations for continuous variables and numbers (percentages) for categorical variables. The associations of the exposome score and its individual scores (the diet score, the activity score, the air pollution score, the sleep score, and the socioeconomic status score) with serum metabolites over eight years were analysed using linear mixed-effects models adjusted for time, age, sex, pubertal status, and serum cotinine, with participant included as random effect (intercept), using the *lmer* package of the R software[123]. Model outputs were inspected to check that the model assumptions were satisfied, and if necessary, metabolite abundancies were log-transformed if it improved model accuracy. We also analysed whether BMI-SDS mediated the associations of the exposome score with serum metabolites over eight years by using the Baron & Kenny method[124]. Moreover, we analysed whether BMI-SDS modified these associations over eight years by adding BMI-SDS and the interaction term for BMI-SDS and the exposome score into the linear mixed-effects models. The *interactions* R package was used to plot significant interactions[125]. A leave-one-out sensitivity analysis was conducted to explore if a specific exposure category was key a contributor to the exposome score. This was done by excluding a single exposure component from the original exposome score. This was repeated for each exposure component, resulting in a total of five "leave-out" models. A sensitivity analysis was also carried out to explore the impact of using different diet scores as part of the formulation of the exposome score—this sensitivity analysis compared the Mediterranean Diet Score and the Baltic Sea Diet Score with the Finnish Children Healthy Eating Index. A graphical representation of the relationships between the exposome score, serum metabolites measured by LC-MS and NMR, and BMI-SDS is displayed in Fig. 2. A *p*-value of <0.05 for a two-tailed test was used to indicate statistical significance, and 95% confidence intervals were inspected to determine if they included the null effect. *P*-values were also corrected for multiple testing, with a false discovery rate of <0.1 deemed significant.

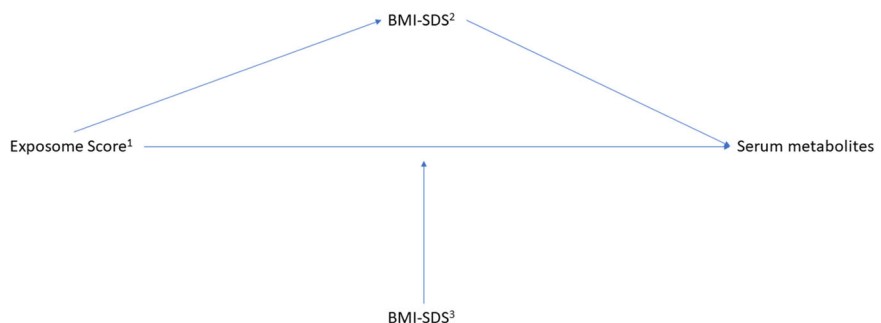

**Fig. 2 | Simplified diagram illustrating potential relationships between the exposome score, serum metabolites and BMI-SDS.** [1]The exposome score will be assessed to determine if it is associated with serum metabolites, and these will be compared to associations of the individual exposure categories with serum metabolites. [2]BMI-SDS will be investigated as a potential mediator of the association of the exposome score on serum metabolites. [3]The association of the exposome score with serum metabolites will be explored to determine if it is modified by BMI-SDS. The exposome score (ranging between 5 and 20) was computed by summing up quartile scores for diet quality (measured by Finnish children healthy eating index), activity (combined score for total physical activity and sedentary time), pollution (combined score for nitrogen oxide, nitrogen dioxide, particulate matter ≤10 μm, particulate matter ≤2.5 μm and ozone), sleep (measured by sleep duration) and socioeconomic status (combined score for household income and parental education). Serum metabolites were measured by liquid chromatography–mass spectrometry and nuclear magnetic resonance. BMI-SDS body mass index standard deviation score.

## Reporting summary

Further information on research design is available in the Nature Portfolio Reporting Summary linked to this article.

## Data availability

Information about the PANIC study and the variables used in the present paper are described at www.panicstudy.fi/en/etusivu. The data are not publicly available due to research ethical reasons and because the owner of the data is the University of Eastern Finland and not the research group. However, the principal investigator of the PANIC study can provide further information on the study and its data on a reasonable request (contact: https://uefconnect.uef.fi/en/person/timo.lakka/).

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

## Acknowledgements

We are indebted to all members of the PANIC research team for their invaluable contribution in the acquisition of the data throughout the study, as well as to all children and their parents participating in the PANIC study. The authors would also like to thank Biocenter Finland (www.biocenter.fi) and Biocenter Kuopio (www.uef.fi/web/bck) for supporting the study. The PANIC Study has received funding from Ministry of Education and Culture of Finland, Ministry of Social Affairs and Health of Finland, Academy of Finland, Research Committee of the Kuopio University Hospital Catchment Area (State Research Funding), Finnish Innovation Fund Sitra, Social Insurance Institution of Finland, Finnish Cultural Foundation, Foundation for Pediatric Research, Diabetes Research Foundation in Finland, Finnish Foundation for Cardiovascular Research, Juho Vainio Foundation, Paavo Nurmi Foundation, Yrjö Jahnsson Foundation, and the city of Kuopio. Moreover, this project has received funding from the European Union's Horizon 2020 research and innovation programme under grant agreement No. 874739.

## Author contributions

D.R.H., M.K. and T.A.L. conceptualised the study. T.A.L., M.K., I.Z., S.M., K.H., E.A.H., A.V. and S.S. provided scientific advice for completion of the study. I.Z. analysed the metabolomics data. S.A. and K.H. developed and supervised the LC-MS method used in this study. D.R.H., I.Z. and S.M. planned and D.R.H. performed the statistical analyses. D.R.H., I.Z., T.A.L., M.K. and S.M. interpreted the results. D.R.H. drafted the manuscript. All authors critically revised the manuscript for its intellectual content and approved the final version of the manuscript. T.A.L. is the principal investigator of the PANIC study.

## Competing interests

Kati Hanhineva is affiliated with Afekta Technologies Ltd. The remaining authors declare that there are no competing interests.
