## [Peer Review File · Communications Biology]

Reviewers' comments:

Reviewer #1 (Remarks to the Author):

1. The concept of a summary index (composite) exposome score is an interesting one, and one that many are not so familiar with. How was this formulated, and what are the major components of the exposome score – this information is available in the method section but should be better summarised in the introduction section. What is the temporal variability of the exposome score within study-participants? Could the authors have demonstrated the intra-class correlation coefficients using the followed-up samples?
2. Have metabolite associations with the exposome score (P-values) been adjusted for multiple testing? Have the authors directly examined whether the change in exposome score is associated with the changes in the metabolite levels in the followed-up samples?
3. Phosphocholine is an intermediate in the synthesis of phosphatidylcholine, while phosphatidylcholine is a compound class, phosphocholine has a well-defined structure with a defined compound mass. Phosphocholine and phosphatidylcholine should not be used interchangeably, and I think the references of phosphocholine in the text and table are incorrect.
4. I cannot find any descriptions of the LCMS data processing and annotation in the Method – this should be directly reported, and/or clearly referenced to a published and peer-reviewed article.
5. The supplementary tables were incorrectly formatted / converted to PDF. Data in table columns and legends were missing or are incomplete. As a result, I have not been able to take account of the information provided in this supplementary table in this review. Please reformat the table appropriately, and if necessary, provide the information in spreadsheet(s).
6. Given that study appear to have identified BMI as an important cofounder / effect modifier in the association analyses (although it is hard to assess the strength of evidence since the supplementary material was not accessible in the version provided in the review), could the authors have stratified their analyses by BMI-SDS?

Reviewer #2 (Remarks to the Author):

In this study, researchers developed an "exposome score" that combines environmental and lifestyle exposures, such as diet quality, physical activity, sedentary time, sleep duration, air pollution, and socioeconomic status. They investigated the associations between this exposome score and serum metabolites measured using LC-MS and NMR in a general population of children. They used linear mix effect models. The exposome score was found to be linked to a wide range of serum metabolites, primarily including phospholipids, fatty acids, amino acids, fatty amides, xenobiotics, and energy-related metabolites. The exposome score showed associations with several of these metabolites, emphasizing the importance of using a composite score to predict metabolic changes linked to various environmental and lifestyle exposures since childhood.

Overall, the manuscript presents an interesting investigation of various factors related to health and

exposure. Addressing these additional points and providing more explanations would enhance the manuscript's clarity and contribute to a deeper understanding of the research conducted.

Please find attached my comments.

****Lines 399-400:**** It might be interesting for the reader to know how different diet qualities (eg UPF, KIDMED) could potentially affect the developed indicators and if this could influence the interpretation of the results.

****Line 415:**** Could you kindly provide any reference or analysis that supports the statement about "rather homogeneous" air pollution in Kuopio? Additionally, is there any association between air pollution and green spaces or rural networks in Kuopio?

****Line 430:**** Considering the wide range of environmental and lifestyle exposure factors explored in the literature (e.g., noise, ambient temperature, green spaces, etc.), it would be interesting to know about the rationale behind the authors' selection of these specific risk factors and their direct association with the study's outcomes. Additionally, I noticed that the PANIC study includes variables related to early-life exposure, such as gestation age and rapid growth. I wonder if these variables could provide any additional insight into the analysis. Why these variables have not been selected in the score?

****Lines 389 to 391:**** Several studies have examined the association of blood metabolomics with different types of diets, such as the Baltic diet, Kid-Med, and malnutrition, which have investigated the effects on metabolomic profiles. Additionally, I noticed that the reference "Zarei et al., submitted" is currently unavailable for further information. Hence, it would be great to mention how existing studies and reviews could provide valuable insights into the development of the presented exposure score and how the developed score's sensitivity and robustness could be explored using other metabolomic traits associated with exposure in the literature.

****Line 459:**** The sensitivity analysis is a valuable tool for assessing the robustness and reliability of scientific findings by testing how changes in different parameters or variables affect study results. In the context of exposure science, I am particularly intrigued by the potential effects of different urban structure characteristics or pollution levels on the models.

****Line 463:**** It is great that the authors present an analysis for the "individual score". Is this part of a sensitivity analysis to investigate the robustness of the exposure score? In this context, it would be highly beneficial if the authors could conduct additional analyses to further explore the development of the score and examine its robustness by applying sensitivity analysis for different variables. Such an approach would add valuable insights to the study and enhance the reliability of the exposure score's findings.

****Line 465:**** Regarding the statistical methods applied, I am wondering if the linear mixed-effect

models refer to a single exposure (wide association approach) or if they account for multiple exposures. Given the importance of controlling false positive errors in scientific research, I would like to inquire if any p-value correction was employed, and if so, whether there were any changes in the significant associations after applying this correction (eg false discovery rate - FDR).

****Line 469:**** The authors mentioned that they checked whether BMI-SDS mediated the associations of the exposome score with serum metabolites by adding BMI-SDS into the model. To investigate the mediation of a variable, I would suggest applying mediation analysis to explore the indirect total effect of BMI-SDS in the model.

****Line 469:**** Regarding child-specific characteristics and unique exposure routes: Measuring and monitoring exposome in childhood can indeed present various challenges. For example, higher breathing rates and surface areas for absorption (relative to body weight) could result in higher internal exposure via inhalation. Exposure patterns, such as spending more time indoors or outdoors, may also influence inhalation. Additionally, children's behaviors, like playing on grass or ground and sleeping longer in bed, could lead to higher magnitude, duration, or frequency of exposure via dermal routes. I think that it would be worth discussing in the paper if there are any challenges when creating an exposure score specifically for children, considering these unique factors that may impact their exposure levels and susceptibility.

****Lines 251-252:**** Organophosphates can be found in PM_{2.5}, but human exposure to organophosphates can also occur through diet. I noticed that in the S3 Comparison of Scores, LC-MS did not show an association of sarcosine with air pollution, but there was an association with activity and socioeconomic status. I would appreciate further explanation and interpretation of these results.

****Line 308:**** Including creatinine in the set of metabolome raises concerns about potential bias in the analyses, particularly with respect to renal failure. To address this, it might be helpful to consider excluding cases with abnormal creatinine blood levels and estimated clearance (GFR) from the model, adjusting for this covariate, or conducting a sensitivity analysis to assess its impact.

****Lines 213 and 214:**** It would be beneficial to know if there are any systematic reviews supporting the evidence presented in these sections, particularly regarding child-specific exposures that occur during childhood and the challenges in measuring them.

****Lastly,**** over the last 20 years, a significant amount of research has been conducted on metabolic and childhood cardiometabolic health, particularly regarding obesity, diabetes, etc. I am curious if there are any systematic reviews that summarize the direction of these associations and if they are consistent across different metabolic platforms. This could provide valuable context for the current study's findings.

minor

****Line 51:**** Ultra-processed foods (UPFs) constitute a category of food products that undergo extensive

industrial processing, often involving the combination of numerous ingredients and additives to create convenient, ready-to-eat, or ready-to-heat products. These foods are typically low in essential nutrients and high in unhealthy components such as salt, added sugars, and unhealthy fats. Consequently, exposure to such foods could have a significant impact on cardio-metabolic health and may affect the metabolic profile. It would be great if the authors could provide a couple of references for the Ultra Process Food Diet (UPF) score and its association with Cardio/Metabolic/Metabolomics health/profiles.

****Lines 78 - 81:**** It might be more appropriate to move this discussion to a later section to ensure a logical flow in the manuscript.

****** A figure with an overview of the proposed approach would be very helpful.

Reviewer #3 (Remarks to the Author):

The study explored the effect of exposome score on serum metabolites to assess the impact of multiple exposures on cardiometabolic risk factors. Overall, the manuscript is well written and clinically meaningful. However, my major concerns are about the accuracy of using summed scoring methods and whole study didn't adjust for multiple testing.

MAJOR COMMENTS

INTRODUCTION:

- Line 53-56. It would be beneficial to also mention the benefit of using sum scores over traditional methods or mixture methods for assessing the effect of multiple exposures.

METHODS:

- For methods on metabolomics analyses, are there any quality assurance and quality control processes on untargeted metabolites before analysis?
- The author mentioned "The metabolites used in the present analyses are those that were previously found to be affected by the physical activity and dietary intervention from baseline to 8-year-follow-up" in another study. Why only chose previously affected metabolites instead of overall metabolites?
- For the assessment of air pollution, during the 8 year of follow-up, did authors consider for the movement of study subjects?
- Line 465, In model strategies, how to deal with the potential medical conditions during the study follow up periods? Especially for those related to cardiometabolic conditions, it can also affect the study results.
- In statistical analysis, I would suggest conduct multiple comparison correction for all results as to reduce the false positive discovery induced by large metabolomics data.

RESULTS:

- For table1, suggest to add one column of p-value for chi-square or ANOVA test between three groups. Also, add unit for cotinine concentrations.
- For table 2, what are metabolite abundances? Are these metabolite intensities? Suggest to clarify.
- For all supplement materials, they can't be shown completely in pdf file. Need to remake new one.

- For figure2, why some y-axis showed metabolite abundances and some showed concentration? Suggest to clarify. Also, panel B seems decrease over time and panel E seems increase, which is opposed to the statement in note. Please double check.

- Overall, the results lack the comparison of significant metabolites between three different models (original, mediator and modification) for LCMS and NMR, respectively. Suggest to visualize them (eg: Venn diagram)

DISCUSSION

- Line 209, the author mentioned: “many of these metabolites observed to be decreased in individuals with a higher exposome score, indicative of a higher overall exposure to harmful environmental and lifestyle factors.” Does this mean that lower levels of these metabolites indicate “bad”? It’s not always the case for pro-inflammatory metabolites.

- Can author also briefly discuss the discrepancy of the direction of associations between LCMS and NMR? For example, line 209-211; line 226-228.

- Suggest to briefly mention why LCMS and NMR these two methods are applied? Advantages or disadvantages, and how they complement with each other?

- As mentioned in limitation, I’m still concerned about the performance of scoring approach as this may oversimplify the complexities of these exposures which are non-linear and have interactive relationships between each other. For example, air pollution can affect physical activity, SES can affect diet choice etc. These are more like to be synergistic. Perhaps consider some sensitivity analysis, like leave out analysis- remove one exposure at a time.

- One more point in limitation, the large percentage of loss to follow up during periods can introduce bias to study.

MINOR COMMENTS

- Line 38, suggest to add SES in these factors as the authors also considered SES in analysis.

- Line 40-42, “However, environmental and lifestyle exposures do not exert their physiological effects in isolation. Instead, there is complex interplay between external exposures, and with their associated internal physiological responses.” Needs reference to support this statement.

- Line 378, the overall study design seems like a cohort study instead of clinical trial in current analysis. What’s the intervention for intervention group? Is intervention/ control group related to current study? If not, suggest to delete.

- Line 394, suggest to change “second-hand smoking” to “environmental exposure to smoking” as third-hand smoking can also be an underlying source for cotinine.

Reviewer #1 (Remarks to the Author):

To the reviewer: Thank you for your insights, especially regarding fixing the compound name in Comment #3. We would like to highlight that, during the completion of the extra analyses for the resubmission, an issue was noted in the original calculation of the quartiles for one of the exposure components. This was rectified, which led to some associations transitioning slightly above/below the significance threshold, which is why there appear to be some changes to the results section of the manuscript.

Comment	Response
1. The concept of a summary index (composite) exposome score is an interesting one, and one that many are not so familiar with. How was this formulated, and what are the major components of the exposome score – this information is available in the method section but should be better summarised in the introduction section. What is the temporal variability of the exposome score within study-participants? Could the authors have demonstrated the intra-class correlation coefficients using the followed-up samples?	Thank you for the positive feedback. We appreciate the suggestion on summarizing the exposome score in the introduction section and have briefly elaborated on the exposome score (LINE 80-84). “In this study, we aimed to compare the outcomes of investigating the combined effect of multiple exposures versus individual exposure categories. To do this, we calculated the magnitude of exposure to diet, activity, sleep, air pollution and socioeconomic status. These individual scores were summed together to create a combined environmental and lifestyle exposure (“exposome”) score.” The temporal variability of the exposome score is presented in Table 1 (PAGE 48), which shows that the mean value of the exposome score remained relatively constant throughout the study. Intra-class correlation coefficients (ICC) can be useful to gain some insight into the variability of the score, and it is a valid suggestion. It was considered due to the large number of dropouts across the three timepoints coupled with missing values might lead to a “messy” dataset and so estimates of variance would be less consistent. As such, it was thought that ICC would not provide much further detail beyond what was already gained in the current analysis, and we did not want to overload the readers with additional methods. However, if the reviewer/editor strongly believe that the addition of ICCs would provide added value, we are happy to incorporate it.
2. Have metabolite associations with the exposome score (P-values) been adjusted for multiple testing? Have the authors directly examined whether the change in exposome score is associated with the changes in the metabolite levels in the followed-up samples?	Thank you for your question. The p-values presented in the manuscript are unadjusted p-values. Due to the low concentrations of pollutants, and the large number of tests associated with metabolomics studies, it was considered that the effect sizes of the pollutants would be quite weak and presenting adjusted p-values would run the risk of completely diluting significance. The supplementary material has been updated to include adjusted p-values. This study was a longitudinal analysis, with the changes of the exposome score and metabolite abundances over time evaluated in the models presented in the manuscript. The relationship between the exposome score and the metabolites at each individual timepoint were not assessed cross-sectionally.
3. Phosphocholine is an intermediate in the synthesis of phosphatidylcholine, while phosphatidylcholine is a compound class, phosphocholine has a well-defined structure with a defined compound mass. Phosphocholine and phosphatidylcholine should not be used interchangeably, and I think the references of phosphocholine in the text and table are incorrect.	Thank you for highlighting this mistake, we have now corrected it throughout the manuscript (Results & Discussion sections, and in the Tables).

4. I cannot find any descriptions of the LCMS data processing and annotation in the Method – this should be directly reported, and/or clearly referenced to a published and peer-reviewed article.	Thank you for this comment. We have now provided further information on the metabolomic analyses and clarified the references for the data processing and metabolite annotations (see LINES 597-633). “Samples were analysed utilising four LC-MS methods. Reversed-phase analyses were performed using a ultra-high performance liquid chromatography (UHPLC)-Orbitrap-mass spectrometry system (Thermo Fischer Scientific, Bremen, Germany), which consisted of a Vanquish UHPLC system using Zorbax Eclipse XDB-C18 column (particle size 1.8 µm, 2.1 × 100 mm; Agilent Technologies), and a Q Exactive Focus mass spectrometer. The hydrophilic interaction liquid chromatography was performed with a UPLC-quadrupole time-of-flight (QTOF)-mass spectrometry system (Agilent Technologies, Santa Clara, CA, USA), which consisted of a 1290 UPLC using Acquity UPLC BEH amide column (2.1 × 100 mm, 1.7 µm; Waters Corp., Milford, MA, USA), and a 6540 UHD QTOF mass spectrometer. Both positive and negative electrospray ionisation were employed, yielding four data files for each sample (Zarei et al., submitted)¹¹¹. After obtaining the molecular features using MS-DIAL v4.90 software, where peak picking and alignment are carried out,¹¹² the data were preprocessed using R software (R Core Team, 2021, https://www.R-project.org) and the notame package.¹¹¹ Data from each of the four analytical modes were processed separately, which included correcting the molecular features for the drift pattern, checking feature quality based on the quality control samples, and imputing the missing values with random forest imputation (Zarei et al., submitted).¹¹¹ Once the preprocessing had been completed, the data was used in statistical analyses, to focus on the molecular features observed to be significantly affected by the physical activity and dietary intervention from baseline to 8-year follow-up examinations in the PANIC study (Zarei et al., submitted), with these compounds selected for identification. Identification of the compounds followed the 4-level annotation scheme proposed by the Metabolomics Standards Initiative,¹¹³ with annotation of the molecular features based on ms/ms spectral comparison on databases including METLIN, MassBank of North America, Human Metabolome Database and LIPID MAPS, as well as published literature. Level 1 identification was based on retention time and ms/ms fragmentation match against in-house purified standards. The metabolites addressed in the present analyses are thus those that were found to be affected by the physical activity and dietary intervention from baseline to 8-year follow-up examinations in the PANIC study (Zarei et al., submitted).”
5. The supplementary tables were incorrectly formatted / converted to PDF. Data in table columns and legends were missing or are incomplete. As a result, I have not been able to take account of the information provided in this supplementary table in this review. Please reformat the table appropriately, and if necessary, provide the information in spreadsheet(s).	Thank you for pointing this out. It is unfortunate that the information in the supplemental table could not be easily viewed. The manuscript was submitted using a “transfer service”, and it appears there were issues with the automatic formatting of the supplementary material. The information was originally provided in an excel file, and seemingly was converted to PDF format.
6. Given that study appear to have identified BMI as an important cofounder / effect modifier in the association analyses (although it is hard to assess the strength of evidence	Thank you for raising this suggestion. Stratifying results is certainly one approach to take when exploring the role of BMI in our study. However, splitting the results in this way would reduce the power in the analyses. Furthermore, using stratification requires adopting some cut-off point. It is possible to use a cut-off such as that separating “normal weight” and “overweight” using the paediatric cut-offs proposed by IOTF (Cole & Lobstein,

since the supplementary material was not accessible in the version provided in the review), could the authors have stratified their analyses by BMI-SDS?	2012). However, this then introduces an artifact of a difference in effect of a BMI-SDS differentiating children with normal weight, overweight or obesity. Investigating the interaction between adiposity and the exposome score in the manner we have allows us to make use of as much data as possible in the analyses, while also allowing for the analysis of BMI as a continuous variable without introducing some artifact associated with the choice of some cut-off value. Cole & Lobstein, 2012: doi:10.1111/j.2047-6310.2012.00064.x
---	--

Reviewer #2 (Remarks to the Author):

In this study, researchers developed an "exposome score" that combines environmental and lifestyle exposures, such as diet quality, physical activity, sedentary time, sleep duration, air pollution, and socioeconomic status. They investigated the associations between this exposome score and serum metabolites measured using LC-MS and NMR in a general population of children. They used linear mix effect models. The exposome score was found to be linked to a wide range of serum metabolites, primarily including phospholipids, fatty acids, amino acids, fatty amides, xenobiotics, and energy-related metabolites. The exposome score showed associations with several of these metabolites, emphasizing the importance of using a composite score to predict metabolic changes linked to various environmental and lifestyle exposures since childhood.

Overall, the manuscript presents an interesting investigation of various factors related to health and exposure. Addressing these additional points and providing more explanations would enhance the manuscript's clarity and contribute to a deeper understanding of the research conducted.

To the reviewer: Thank you for your insights, we really appreciate the thoroughness in which you considered the manuscript and are grateful for the feedback which we feel has allowed us to greatly enhance the quality of the manuscript. We would like to highlight that, during the completion of the extra analyses for the resubmission, an issue was noted in the original calculation of the quartiles for one of the exposure components. This was rectified, which lead to some associations transitioning slightly above/below the significance threshold, which is why there appear to be some changes to the results section of the manuscript.

Comment	Response
1. **Lines 399-400:** It might be interesting for the reader to know how different diet qualities (eg UPF, KIDMED) could potentially affect the developed indicators and if this could influence the interpretation of the results.	Thank you for this suggestion. We have since conducted a sensitivity analysis testing different measures of diet quality in the computation of the exposome score to explore the impact this has on results (LINES 273-285) and it has been expanded upon in the discussion (LINES 487-512). Results: "Comparison of diet scores within exposome score The use of the Finnish Children Healthy Eating Index, Mediterranean Diet Score, or Baltic Sea Diet Score in the formulation of the exposome score was significantly associated with 16, 27 and 20, respectively, serum metabolites measured by LC-MS (Supplementary Table S3). Altogether, 11 metabolites were common to all three models. However, there were four, eight and two unique associations when the Finnish Children Healthy Eating Index, the Mediterranean Diet Score, and the Baltic Sea Diet Score were used in the exposome score, respectively (Supplementary Figure S3). The use of the Finnish Children Healthy Eating Index, the Mediterranean Diet Score, or the Baltic Sea Diet Score in the formulation of the exposome score was significantly associated with 15, 11 and 22, respectively, serum metabolites measured by NMR (Supplementary Table S4). Four metabolites were common to all three models. However, there were five, two and seven unique associations when the Finnish Children Healthy Eating Index, the Mediterranean Diet Score, and the Baltic Sea Diet Score were used in the exposome score, respectively (Supplementary Figure S4)."

	Discussion: “To demonstrate the challenges of developing a composite exposure score and exploring its feasibility, we conducted a sensitivity analysis using just one component of the exposome score (diet), where the use of different, validated, approaches to assess diet quality were compared. The original exposome score model, using the Finnish Children Healthy Eating Index, was compared to two sensitivity exposome models using either the Baltic Sea Diet Score or the Mediterranean Diet Score. While there was some overlap between the three models, there were also a number of unique associations with serum metabolites measured by LC-MS and NMR based on the diet score incorporate into the exposome score (Supplementary Figures S3-S4). Interestingly, when we compared the results of the sensitivity analyses using the Finnish Children Healthy Eating Index and the Baltic Sea Diet Score, both of which have been validated in a Finnish population,^{20,21} we found that almost a third of the metabolites measured by LC-MS and NMR that were associated with the original exposome score were not found to be related to the exposome score with the Baltic Sea Diet Score. Future research should consider the suitability of selected measures to the population studied, and further research is necessary to determine optimal approaches to assess the impact of diet on an individuals’ overall exposome, whether this be through a diet quality approach, or the assessment of specific foods or nutrients. Future research should also explore the impact of ultra-processed foods, which have been positively associated with obesity and other cardiometabolic risk factors in children and adolescents.¹⁰⁵ Specifically, animal-based products and sugar-sweetened beverages should be examined, which were recently determined as the main subgroups of ultra-processed foods associated with cardiometabolic multimorbidity in adults across 10 European countries.¹⁰⁶ These are insights based on the diet component of the exposome score, with special care likely needed for all components incorporated into a composite exposure score. This highlights the need to carefully reflect upon and select appropriate measures, and thus researchers should be careful in the planning of composite scores to measure multiple exposures, and the approaches used to assess the components to be incorporated.”
2. **LINE 415:** Could you kindly provide any reference or analysis that supports the statement about "rather homogeneous" air pollution in Kuopio? Additionally, is there any association between air pollution and green spaces or rural networks in Kuopio?	The statement relies on unpublished work done by Dr. Mikkonen, who is a coauthor in this study, and the Aerosol Physics Research group at the University of Eastern Finland. They have compared the pollutant concentrations between multiple measurement sites within the city area and found that only by the motorway were the concentrations significantly higher than in other parts of the city, but the particulate pollution or noise were not affecting the exposure levels in residential areas. The air quality network data are not yet published, but there are multiple papers introducing measurements from Puijo tower measurement site, which is close to the city, such as (Portin et al., 2014). They show that the overall levels in Kuopio are low. To the best of our knowledge, there is no study reporting on associations of green spaces with air pollution in Kuopio. However, as detailed in the following response to your comment on LINE 430, the majority of residents in Kuopio live within close proximity to green and blue spaces, while air pollution was considered to be relatively homogenous in Kuopio. Portin et al., 2014: https://doi.org/10.5194/acp-14-6021-2014
3. **Line 430:** Considering the wide range of environmental and lifestyle exposure factors explored in the literature (e.g., noise, ambient temperature,	Thank you for your question, the comment on early-life exposures is especially valid. The exposure variables that were ultimately used were selected based on data available within the study. Unfortunately, participant addresses were not available in this study and so it was not possible to use modelled data for urbanization, green spaces etc. Then, based on the selected variables, they

green spaces, etc.), it would be interesting to know about the rationale behind the authors' selection of these specific risk factors and their direct association with the study's outcomes. Additionally, I noticed that the PANIC study includes variables related to early-life exposure, such as gestation age and rapid growth. I wonder if these variables could provide any additional insight into the analysis. Why these variables have not been selected in the score?	were grouped based on similar categories (such as the different air pollutants being integrated into a pollution score). Measured data for noise were not available and the modelled noise data are not applicable in Kuopio. As for green (and blue) spaces; Kuopio has many lakes, forests, parks etc. heavily interspersed throughout the region. (Viinikka et al., 2023) observed that 95% of residents in Kuopio lived within 300 metres of a green area, and that 89% or greater of residents lived within 1000 metres of water bodies, green areas with routes, forests, and large green areas. Temperature is an interesting variable to consider. Temperatures in Kuopio can change dramatically, easily reaching -15°C and even colder in winter and 25°C and hotter in summer. There are negative health effects in both ranges of the scale, which have not been as well-researched as adverse health effects associated with exposure to air pollution. Thus, the impact of temperature, especially with cold extremes, was left to subsequent studies to allow for these effects to be studied and understood with a greater emphasis. Early-life exposures were initially considered to be incorporated into the score. However, there were some issues with data missingness with some of these variables (such as maternal BMI prior to pregnancy and gestational weight gain). We have now added this to the limitations to acknowledge the role of early-life exposures and how it was not accounted for in this study (LINES 554-556) and should be considered in future research. Thus, the role of early life exposure will be studied in a separate study, using a subset of the participants who have valid early-life measurements. “Due to issues with data missingness, it was not possible to explore the role of early-life exposures, and subsequent studies should consider early-life exposures, such as maternal exposures during pregnancy, in the context of the exposome.” Viinikka et al., 2023: https://doi.org/10.1016/j.apgeog.2023.102973
4. **Lines 389 to 391.** Several studies have examined the association of blood metabolomics with different types of diets, such as the Baltic diet, Kid-Med, and malnutrition, which have investigated the effects on metabolomic profiles. Additionally, I noticed that the reference "Zarei et al., submitted" is currently unavailable for further information. Hence, it would be great to mention how existing studies and reviews could provide valuable insights into the development of the presented exposure score and how the developed score's sensitivity and robustness could be explored using other metabolomic traits associated with exposure in the literature.	Thank you for this suggestion. We have expanded upon the use of metabolomics in the introduction, specifically introducing reviews that support the use of metabolomics in exposure research (LINES 99-105). “The application of metabolomics in understanding the exposome has already been proposed,²⁸ while metabolomics has been used in exposure research, with numerous reviews supporting the notion of a metabolic response with exposure to air pollution^{29,30}, physical activity^{31,32}, diet^{33,34}, and sleep^{35,36}, while it has also been demonstrated that socioeconomic status can affect the human metabolome.³⁷ This is further supported with findings from a systematic review suggesting that combined healthy lifestyle behaviours are associated with a unique metabolic profile.³⁸” Our research question focused on the idea of how the exposome can affect cardiometabolic health and using metabolomics to better understand the pathways/mechanisms through which combined exposures may be associated with adverse health outcomes. We have not focused on specific metabolomic profiles associated with “different exposomes”, and so comparing profiles with the literature is a little out of the scope of this paper. Future research, whether through the use of sum scores, exposome-wide association studies, or some other composite measure, could consider different exposomes and associated metabolomic profiles, and compare with the literature, such as with the systematic review by Kaspy et al., to see if the findings are supported. This has now been added as an aspect of future research worth considering (LINES 570-572).

	“We have also investigated associations with individual metabolites, and it may be worthwhile to explore metabolomic profiles attributed to different exposome patterns.” Kaspy et al., 2022: https://doi.org/10.1002/pmic.202100388
5. **Line 459:** The sensitivity analysis is a valuable tool for assessing the robustness and reliability of scientific findings by testing how changes in different parameters or variables affect study results. In the context of exposure science, I am particularly intrigued by the potential effects of different urban structure characteristics or pollution levels on the models.	This is a valid and really good point. Unfortunately, we did not have data on participant addresses and so it was not possible to use various data like urbanization maps, while questionnaires data on topics such as “distance to [green space]” was not measured in the study. It would indeed be great to investigate other environmental exposures such as greenness (and blue spaces) and urbanization. The geography of Kuopio differs from large European cities such as Paris or Amsterdam, with Kuopio surrounded by nature, including lakes and parks/forests. As mentioned in the response to Comment 3, (Viinikka et al., 2023) observed that 95% of residents in Kuopio lived within 300 metres of a green area, and that 89% or greater of residents lived within 1000 metres of water bodies, green areas with routes, forests, and large green areas. The primary aim of this current study was somewhat proof-of-concept, to explore the need for researchers to consider and assess the combined impact of multiple exposures, instead of the standard biomedical approach which typically looks at single exposures in isolation. We are planning follow-up analyses with a greater methodological focus once participants’ addresses have been obtained. This will allow the use of modelled data maps, such as for green and blue spaces and urbanization. We have also performed further sensitivity analyses, such as a leave-out analysis as well as computing the exposome score using various diet scores, to show how using different approaches for just one of the five exposure categories can influence the findings (LINES 273-303; 481-531). Results: “Comparison of diet scores within exposome score The use of the Finnish Children Healthy Eating Index, Mediterranean Diet Score, or Baltic Sea Diet Score in the formulation of the exposome score was significantly associated with 16, 27 and 20, respectively, serum metabolites measured by LC-MS (Supplementary Table S3). Altogether, 11 metabolites were common to all three models. However, there were four, eight and two unique associations when the Finnish Children Healthy Eating Index, the Mediterranean Diet Score, and the Baltic Sea Diet Score were used in the exposome score, respectively (Supplementary Figure S3). The use of the Finnish Children Healthy Eating Index, the Mediterranean Diet Score, or the Baltic Sea Diet Score in the formulation of the exposome score was significantly associated with 15, 11 and 22, respectively, serum metabolites measured by NMR (Supplementary Table S4). Four metabolites were common to all three models. However, there were five, two and seven unique associations when the Finnish Children Healthy Eating Index, the Mediterranean Diet Score, and the Baltic Sea Diet Score were used in the exposome score, respectively (Supplementary Figure S4). Leave-one-out analysis The original exposome score model and leave-out models were associated with 54 serum metabolites measured by LC-MS (Supplementary Table S7). The directions of the significant associations of the exposome score with serum metabolites measured by LC-MS were consistent across the various leave-out models (Supplementary Table S7). Most of the models were observed to have a

similar number of significant associations with metabolites as the original exposome score model, ranging between 13 and 16 (Supplementary Figure S1). The deviation to this trend was found with the removal of the air pollution category from the exposome score; this model had 29 significant associations with metabolites, which was the largest number of such associations (Supplementary Figure S1, Supplementary Table S7).

The original exposome score model and leave-out models were associated with 27 serum metabolites measured by NMR (Supplementary Table S8). The directions of the significant associations of the exposome score with serum metabolites measured by NMR were consistent across the various leave-out models (Supplementary Table S8). In comparison with the original exposome score model, the leave-out models for diet, activity, sleep, and socioeconomic status were all significantly associated with less metabolites (8, 5, 12 and 8, respectively). The exception to this was the leave-out-model for pollution, which was significantly associated with 23 metabolites (Supplementary Figure S2, Supplementary Table S8)."

Discussion:

"Measuring the exposome is a major challenge, and there have been efforts in recent years to measure multiple exposures and try to account for the interactions between them more accurately.¹⁰⁰ One approach that has been implemented in other fields is the use of sum scores, such as in dietary research^{20,21} and the development of various health scores.^{17,18} While composite scores are widely used in research,^{101,102} there are limitations to their use in clinical research,¹⁰² many of which are also relevant to exposome research. Despite the limitations, it is argued that the use of scores allows the net effect of the overall factors to be assessed.^{103,104} To demonstrate the challenges of developing a composite exposure score and exploring its feasibility, we conducted a sensitivity analysis using just one component of the exposome score (diet), where the use of different, validated, approaches to assess diet quality were compared. The original exposome score model, using the Finnish Children Healthy Eating Index, was compared to two sensitivity exposome models using either the Baltic Sea Diet Score or the Mediterranean Diet Score. While there was some overlap between the three models, there were also a number of unique associations with serum metabolites measured by LC-MS and NMR based on the diet score incorporate into the exposome score (Supplementary Figures S3-S4). Interestingly, when we compared the results of the sensitivity analyses using the Finnish Children Healthy Eating Index and the Baltic Sea Diet Score, both of which have been validated in a Finnish population,^{20,21} we found that almost a third of the metabolites measured by LC-MS and NMR that were associated with the original exposome score were not found to be related to the exposome score with the Baltic Sea Diet Score. Future research should consider the suitability of selected measures to the population studied, and further research is necessary to determine optimal approaches to assess the impact of diet on an individuals' overall exposome, whether this be through a diet quality approach, or the assessment of specific foods or nutrients. Future research should also explore the impact of ultra-processed foods, which have been positively associated with obesity and other cardiometabolic risk factors in children and adolescents.¹⁰⁵ Specifically, animal-based products and sugar-sweetened beverages should be examined, which were recently determined as the main subgroups of ultra-processed foods associated with cardiometabolic multimorbidity in adults across 10 European countries.¹⁰⁶ These are insights based on the diet component of the exposome score, with special care likely needed for all components incorporated into a composite exposure score. This highlights the need to carefully reflect upon and select appropriate measures, and thus researchers should be careful in the

	planning of composite scores to measure multiple exposures, and the approaches used to assess the components to be incorporated. Another limitation of the use of composite scores is that it risks over-simplifying the relationships between the various components incorporated into the score. To assess the contribution of the individual exposure components to the exposome score, we conducted a leave-one-out sensitivity analysis. No individual component appeared to be influencing the directions of the associations, with the direction remaining consistent between the exposome score and the various leave-out models. However, it appeared that the air pollution score had a large influence on the number of significant associations, with the leave-out model for pollution significantly associated with the largest number of metabolites measured by both LC-MS and NMR. Interestingly, when we inspect metabolites that were significantly associated with the pollution score and at least one other exposure component, we can see that the directions differ in most instances. It is possible that the pollution score is exerting a small effect in the opposing direction to the other exposures, and while not a large enough effect to be significant, it may be enough to influence the overall exposome score, as evidence by the comparison of the original exposome score with the exposome score from which the air pollution component was removed. The assumption that there is a consistent direction for the various factors incorporated into scores, known as the assumption of uniform directionality, is a disadvantage of a scoring approach.¹⁰⁷ This remains an issue for many of the newer methods that try to measure the mixture effect of various exposures,¹⁰⁰ although there are methods being developed that try to accommodate the opposing directions of different exposures.¹⁰⁸ Viinikka et al., 2023: https://doi.org/10.1016/j.apgeog.2023.102973
6. **Line 463:** It is great that the authors present an analysis for the "individual score". Is this part of a sensitivity analysis to investigate the robustness of the exposure score? In this context, it would be highly beneficial if the authors could conduct additional analyses to further explore the development of the score and examine its robustness by applying sensitivity analysis for different variables. Such an approach would add valuable insights to the study and enhance the reliability of the exposure score's findings.	Thank you for your question. The main aim of the individual score was to highlight the difference in results if associations with an individual exposure were investigated (e.g., diet) compared to associations with the overall combination of multiple exposures, highlighting the potential value for health research to assess the impact of combined exposures, as individuals are not typically exposed to these factors in controlled isolation. We have added further sensitivity analyses, specifically a leave-out analysis, and an analysis that explores the use of different diet scores to measure diet quality, to look at the effect of changing just one of the five exposure categories (LINES 273-303; 481-531; see response to Comment #5 for the specific changes). This is a great suggestion and is something of interest in a follow-up paper that is more methodological in nature in the development of an exposome score to conduct further analyses that incorporate a harder endpoint. Metabolomics is quite variable, and so is used more as a proof-of-concept to test the feasibility and usability of such a composite exposure score. Future analyses will assess different combinations of the various exposure categories to further explore the interactions between each while also investigating the weighting of the various categories based on a hard endpoint, whether that be some sort of incidence or mortality measure, or a marker of disease progression, which can be carried out in adult populations where disease prevalence may be higher.
7. **Line 465:** Regarding the statistical methods applied, I am wondering if the linear mixed-effect models refer to a single exposure (wide association approach) or if they account for multiple exposures. Given the importance of controlling false positive errors in scientific research, I would like to	Thank you for your questions. The mixed-effect models were considered to account for multiple exposures, both through the incorporation of multiple exposures into the exposome score, as well as adjustment for other variables in the models themselves. The p-values presented in the manuscript are unadjusted p-values. Due to the low concentrations of pollutants, and the large number of tests associated with metabolomics studies, it was considered that the effect sizes of the pollutants would be quite weak and presenting adjusted p-values would run the risk of completely diluting significance. The supplementary material has been updated

inquire if any p-value correction was employed, and if so, whether there were any changes in the significant associations after applying this correction (eg false discovery rate - FDR).	to include adjusted p-values. The changes in significant associations after correcting for multiple testing is presented below: P-values: LC-MS Using p value < 0.05 to determine significance, the following numbers of significant associations with serum metabolites measured by LC-MS were observed:  • Exposome score using Finnish Children Healthy Eating Index: 16 • Exposome score using Baltic Sea Diet Score: 20 • Exposome score using Mediterranean Diet Score: 27 • Effect modification by adiposity: 12 After adjusting for multiple tests: using a cutoff of FDR < 0.1 to determine significance:  • Exposome score using Finnish Children Healthy Eating Index: 0 • Exposome score using Baltic Sea Diet Score: 0 • Exposome score using Mediterranean Diet Score: 3 • Effect modification by adiposity: 0 P-values: NMR Using p value < 0.05 to determine significance, the following numbers of significant associations with serum metabolites measured by NMR were observed:  • Exposome score using Finnish Children Healthy Eating Index: 15 • Exposome score using Baltic Sea Diet Score: 22 • Exposome score using Mediterranean Diet Score: 11 • Effect modification by adiposity: 25 After adjusting for multiple tests: using a cutoff of FDR < 0.1 to determine significance:  • Exposome score using Finnish Children Healthy Eating Index: 4 • Exposome score using Baltic Sea Diet Score: 17 • Exposome score using Mediterranean Diet Score: 7 • Effect modification by adiposity: 30
8. **Line 469:** The authors mentioned that they checked whether BMI-SDS mediated the associations of the exposome score with serum metabolites by adding BMI-SDS into the model. To investigate the mediation of a variable, I would suggest applying mediation analysis to explore the indirect total effect of BMI-SDS in the model.	Thank you for highlighting this, which has afforded us the opportunity to communicate more clearly the approach that was used. It is true that BMI-SDS was added to models to explore its role as a mediator. This was done within the framework established by Baron & Kenny, where a series of regression models are used to determine if a variable mediates the association of an exposure with an outcome, under the assumption of mediation. The sentence in statistical methods has been amended to clarify this (LINE 713). “We also analysed whether BMI-SDS mediated the associations of the exposome score with serum metabolites over eight years by using the Baron & Kenny method.¹²³” In our case, the exposome score was positively associated with BMI_SDS (this has also now been specifically stated in the results section, LINE 211-212). BMI-SDS was then explored as a mediator of significant associations of the exposome score with metabolites measured by LC-MS and NMR. “The exposome score was positively associated with BMI-SDS ($\beta = 0.0269$ 95% CI [0.0063, 0.0476, $p = 0.010$].” Future work will consider the role of adiposity when not a mediator, whether it acts in a suppressive manner or is a confounder (the connection between mediation, confounding and suppression has been well-described by MacKinnon et al.).

	Other approaches could also have been used, such as structural equation modelling. It was considered whether the number of data was sufficient when dealing with repeated measures, and so we proceeded with the Baron and Kenny method (MacKinnon, Fairchild & Fritz, 2007). MacKinnon et al., 2000: doi: 10.1023/a:1026595011371 MacKinnon, Fairchild & Fritz, 2007: doi: 10.1146/annurev.psych.58.110405.085542
9. **Line 469:**Regarding child-specific characteristics and unique exposure routes: Measuring and monitoring exposome in childhood can indeed present various challenges. For example, higher breathing rates and surface areas for absorption (relative to body weight) could result in higher internal exposure via inhalation. Exposure patterns, such as spending more time indoors or outdoors, may also influence inhalation. Additionally, children's behaviors, like playing on grass or ground and sleeping longer in bed, could lead to higher magnitude, duration, or frequency of exposure via dermal routes. I think that It would be worth discussing in the paper if there are any challenges when creating an exposure score specifically for children, considering these unique factors that may impact their exposure levels and susceptibility.	Thank you for raising this point, and it is extremely valid. We spent quite a bit of time reflecting on the differences in conducting exposure research in children vs. adults, and ultimately felt that this topic perhaps warrants a review of its own. Future work that may be very beneficial would be a discussion of different exposome variables and how exposures to these variables, and their impacts on health, may differ in children and adults (and also differing between females and males). There is no clear difference in exposure to socioeconomic status or diet between children and adults, but it's possible their impacts on cardiometabolic health may differ. The reviewer has already effectively clarified the differences in exposures to air pollution, and this can be similar for other specific environmental exposures like green spaces. Levels of physical activity are often higher, and levels of sedentary time lower, in children than in adults, however if this activity equates to increased time spent outdoors then there could be increased duration of exposure to other environmental factors. As touched upon, while the exposure to these factors may differ, the physiological responses to these exposures may also differ between children and adults. Due to the scope of this topic, we have resorted to simply acknowledging the difference in assessing exposures in children vs. adults and addressing the need for further research (LINES 572-574). "Furthermore, due to differing exposure patterns as well as the different impacts the various exposures may have on children compared to adults, the use of composite exposure scores in adults still needs to be explored." However, we do agree that this aspect of research is worth considering, and so if the reviewer/editor feel that it is still worthwhile to include in further detail in the discussion and that it does not make the article too long, then we are happy to revisit this discussion point.
10. **Lines 251-252:** Organophosphates can be found in PM2.5, but human exposure to organophosphates can also occur through diet. I noticed that in the S3 Comparison of Scores, LC-MS did not show an association of sarcosine with air pollution, but there was an association with activity and socioeconomic status. I would appreciate further explanation and interpretation of these results.	Thank you for pointing this out. While the statement was valid and supported with references, it was not relevant in the context of the flow of the discussion, which has focused on the association of the exposome score with metabolites, and the potential role/implication of these metabolites with respect to cardiometabolic health. This sentence has been amended to suit this flow better, instead discussing the association of sarcosine with cardiometabolic health (LINES 358-360). "In a recent study, elevated sarcosine was determined to be a marker of dyslipidemia, with the increase in sarcosine even more pronounced across profiles that included further risk factors for metabolic syndrome.⁶³"
11. **Line 308:** Including creatinine in the set of metabolome raises concerns about potential bias in the analyses, particularly with respect	Thank you for raising this concern. The metabolites measured by LC-MS in this study were a subset of metabolites from a previous study (currently under review), namely those that were affected by a lifestyle intervention in children and were amenable for identification. The previous study used untargeted metabolomics, so creatinine was not specifically included in a set of metabolites

to renal failure. To address this, it might be helpful to consider excluding cases with abnormal creatinine blood levels and estimated clearance (GFR) from the model, adjusting for this covariate, or conducting a sensitivity analysis to assess its impact.	but was just part of the overall measured metabolome. In the referred-to LINE 308, we wanted to acknowledge that while creatinine may be a marker of renal and cardiovascular risk, this is observed in adults, and thus the significant association observed in our finding may not be indicative of similar risk. It should also be noted that our study population is a population-based sample of children/adolescents with low prevalence of diseases, and none of them is reported to have a diagnose of renal failure at baseline, 2-year follow-up, or 8-year follow up. Instead, it is considered possible that increases in creatinine might be attributed to the study itself, which incorporated a physical activity element, as growth of children with a higher diet quality and increased physical activity may have contributed to increased muscle mass. Despite all of this, the concern about the potential bias is valid. Running a sensitivity analysis for the original models (association of exposome score with serum metabolites) with adjustment for creatinine, we see the following changes: NMR:  • Main model: 16 significant associations • Adjusting for creatinine: 15 significant associations. • The change is seen with isoleucine, which goes from a p-value = 0.035 to a p-value = 0.0505. All effect directions remain the same. LC-MS:  • Main model: 16 significant associations • Adjusting for creatinine: 16 significant associations. • However, there are a couple of changes in results. After adjusting, we lose a significant association with an o-glycosyl compound with formula C₁₅H₂₈O₁₁ and indolelactic acid, and gain a significant association with dibutyl decanedioate and lysophosphatidylethanolamine 18:1/0:0. All directions of other associations remain the same.
12. **Lines 213 and 214:** It would be beneficial to know if there are any systematic reviews supporting the evidence presented in these sections, particularly regarding child-specific exposures that occur during childhood and the challenges in measuring them.	Thank you for this suggestion. To the best of our knowledge, there are not many systematic reviews on amino acids, lipids, or NMR metabolomics in children and their associations with cardiometabolic outcomes. There are original research articles, which primarily have been included in the discussion section of this manuscript and non-systematic reviews (Frohnert and Rewers, 2016; Mianieri, La Bella and Chiarelli, 2023). There are some systematic reviews on lipoproteins (Ulloque-Badaracco et al., 2023; Palmeira et al., 2013), however we primarily discussed the significant findings in the discussion and the exposome score was not observed to be significantly associated with lipoproteins in our study. The only systematic reviews to cover the metabolites that may underlie cardiometabolic disease pathophysiology were done by Handakas et al., and De Spiegeleer et al., however these both focused on pediatric obesity. A recent systematic review on the association of branched-chain amino acids with cardiometabolic disorders included papers with all age ranges, yet no studies with children were included (Yehia et al., 2023). In some places, where research in children was limited, we have tried to include systematic reviews that were based on findings in adults (LINES 402-405; Li et al., 2022). “In adults, higher levels of circulating saturated fatty acids have been associated with an increased risk of cardiometabolic diseases in a meta-analysis by Li et al.⁸³, whereas monounsaturated fatty acids were positively associated with risk of coronary artery disease in a large prospective cohort study.⁸⁴”

	Many of the systematic reviews and meta-analyses looking at cardiometabolic health in children/adolescents tend to report on the exposures themselves, for example with socioeconomic status (Slopen et al., 2013), diet (Rocha et al., 2017), activity (Verswijveren et al., 2018), air pollution (Yan et al., 2021) and sleep duration (Sun et al., 2020). However, for most of the discussion, we did not discuss individual components of the exposome score. Instead, we looked at associations of the exposome score with serum metabolites measured by LC-MS and NMR, and reported the evidence from the literature on the association of these metabolites with cardiometabolic diseases. Frohnert and Rewers, 2016: doi: 10.1111/pedi.12323 Mianieri, La Bella and Chiarelli, 2023: doi: 10.3390/biomedicines11030809 Ulloque-Badaracco et al., 2023: https://doi.org/10.1186/s12944-023-01860-w Palmeira et al., 2013: doi: 10.1590/S0103-05822013000400017 Handakas et al., 2021: https://doi.org/10.1111/obr.13384 De Spiegeleer et al., 2021: doi: 10.1186/s10020-021-00394-0 Yehia et al., 2023: https://doi.org/10.1093/nutrit/nuac051 Li et al., 2022: doi: 10.3389/fnut.2022.963471 Slopen et al., 2013: https://doi.org/10.1371/journal.pone.0064418 Rocha et al., 2017: DOI: 10.1016/j.jpeds.2017.01.002 Verswijveren et al., 2018: https://doi.org/10.1371/journal.pone.0201947 Yan et al., 2021: https://doi.org/10.1016/j.scitotenv.2021.147279 Sun et al., 2020: https://doi.org/10.1016/j.smr.2020.101338
13. **Lastly,** over the last 20 years, a significant amount of research has been conducted on metabolic and childhood cardiometabolic health, particularly regarding obesity, diabetes, etc. I am curious if there are any systematic reviews that summarize the direction of these associations and if they are consistent across different metabolic platforms. This could provide valuable context for the current study's findings.	Thank you for raising this point. As discussed in the response to Comment 12, to the best of our knowledge, there are limited systematic reviews summarizing the direction, other than two systematic reviews, by Handakas et al., 2021 and De Spiegeleer et al., 2021, however both focused on children with overweight/obesity. Instead, most of the evidence appears to look at the exposures themselves (diet, physical activity, sleep etc., see Comment 12 response). Perhaps this is an identified gap that can be followed up on, to perform a systematic review and meta-analysis of metabolites that are associated with cardiometabolic health in children, especially those which may predict the onset of cardiometabolic diseases later in life. Handakas et al., 2021: https://doi.org/10.1111/obr.13384 De Spiegeleer et al., 2021: doi: 10.1186/s10020-021-00394-0
14. **Line 51:** Ultra-processed foods (UPFs) constitute a category of food products that undergo extensive industrial processing, often involving the combination of numerous ingredients and additives to create convenient, ready-to-eat,	Thank you for this suggestion. It is a very valid point that is raised. It is quite difficult to explain each of the exposure components and all the ways different ways of measuring, different aspects of each exposure etc. can influence the results. However, while the comparison of different diets, levels of processing etc. is a little out of the scope, we have incorporated the suggestion by acknowledging the role food processing may play in influencing the exposome score beyond just diet quality. This is included in the discussion of the sensitivity analysis where the exposome score was computed using different

or ready-to-heat products. These foods are typically low in essential nutrients and high in unhealthy components such as salt, added sugars, and unhealthy fats. Consequently, exposure to such foods could have a significant impact on cardio-metabolic health and may affect the metabolic profile. It would be great if the authors could provide a couple of references for the Ultra Process Food Diet (UPF) score and its association with Cardio/Metabolic/Metabolomics health/profiles.	diet scores to measure diet quality, and future research can adopt a similar analytical approach to that presented in this article to conduct research to target questions such as the role of ultra-processed foods. The specific sentences acknowledging the potential role of food processing is presented in LINES 503-507. “Future research should also explore the impact of ultra-processed foods, which have been positively associated with obesity and other cardiometabolic risk factors in children and adolescents.¹⁰⁵ Specifically, animal-based products and sugar-sweetened beverages should be examined, which were recently determined as the main subgroups of ultra-processed foods associated with cardiometabolic multimorbidity in adults across 10 European countries.¹⁰⁶”
15. **Lines 78 - 81:** It might be more appropriate to move this discussion to a later section to ensure a logical flow in the manuscript.	Thank you for this suggestion. This sentence has now been moved to the opening paragraph of the discussion (LINES 311-313). “To the best of our knowledge, this is the first longitudinal study to investigate the association between an exposome score and metabolic health across childhood and adolescence.”
16. ** A figure with an overview of the proposed approach would be very helpful.	Thank you for this suggestion. A simplified diagram was provided in Figure 1 (PAGE 45). If this is not what was intended, specific suggestions on improving the figure (or visualizing the approach in a different manner) are welcome and can be incorporated in a further revision.

Reviewer #3 (Remarks to the Author):

The study explored the effect of exposome score on serum metabolites to assess the impact of multiple exposures on cardiometabolic risk factors. Overall, the manuscript is well written and clinically meaningful. However, my major concerns are about the accuracy of using summed scoring methods and whole study didn't adjust for multiple testing.

To the reviewer: Thank you for your insights, we really appreciate the thoroughness in which you considered the manuscript and are grateful for the feedback which we feel has allowed us to greatly enhance the quality of the manuscript, especially the comments regarding the statistics and scoring. We would like to highlight that, during the

completion of the extra analyses for the resubmission, an issue was noted in the original calculation of the quartiles for one of the exposure components. This was rectified, which lead to some associations transitioning slightly above/below the significance threshold, which is why there appear to be some changes to the results section of the manuscript.

Comment	Response
1. Line 53-56. It would be beneficial to also mention the benefit of using sum scores over traditional methods or mixture methods for assessing the effect of multiple exposures.	Thank you for this suggestion. We have expanded upon the use of sum scores in the introduction (LINES 54-66) to briefly justify their potential application in exposome research. “However, measuring the exposome can be challenging¹⁵; for example, it can be difficult to account for the sheer number of exposures individuals are exposed to over their life course, the different types of exposures and the tools/technologies able to measure these exposures, and the ability to accurately capture an individual’s personal exposure.^{15,16} One approach used in other areas of research that could be adopted for exposome research is the use of scores. Scores have been used in health sciences to assess cardiometabolic health, with the cardiometabolic risk score for children and adolescents¹⁷ and “Life’s Essential 8” proposed by the American Heart Association.¹⁸ Sum scores have also been used to assess combined lifestyle factors and their impact on health,¹⁹ however the majority of such studies have been carried out in adults. Along with measuring multiple aspects of health, sum scores have also been used to measure diet quality, where the overall impact of the diet is assessed as opposed to investigating single nutrients. Examples of the use of sum scores in nutrition research include the Mediterranean Diet Score²⁰ and the Baltic Sea Diet Score.²¹” We have also discussed the use of sum scores, as well as some advantages and disadvantages, in the discussion when we reflect upon the development of our own exposome score (LINES 483-487, 513-514, 526-531). “One approach that has been implemented in other fields is the use of sum scores, such as in dietary research^{20,21} and the development of various health scores.^{17,18} While composite scores are widely used in research,^{101,102} there are limitations to their use in clinical research,¹⁰² many of which are also relevant to exposome research. Despite the limitations, it is argued that the use of scores allows the net effect of the overall factors to be assessed.^{103,104}” “Another limitation of the use of composite scores is that it risks oversimplifying the relationships between the various components incorporated into the score.” “The assumption that there is a consistent direction for the various factors incorporated into scores, known as the assumption of uniform directionality, is a disadvantage of a scoring approach.¹⁰⁷ This remains an issue for many of the newer methods that try to measure the mixture effect of various exposures,¹⁰⁰ although there are methods being developed that try to accommodate the opposing directions of different exposures.¹⁰⁸”
2. For methods on metabolomics analyses, are there any quality assurance and quality control processes on untargeted metabolites before analysis?	Thank you for your question. The samples in LC-MS metabolomics were measured in five batches, with both within-batch and between-batch QC samples used. A pooled sample comprised of all samples from batch 1 was used as the between-batch QC, while within-batch QCs were achieved by combining a small portion of the batch’s samples. QC samples were included in the analysis at the beginning, to stabilize the instrument and method, while two QCs were included every 12 samples thereafter (one within-batch and one between-batch QC). The QC samples were used for monitoring the

performance of the LC-MS instrument during the analytical run, and they were also used in the data preprocessing for the within- and between-batch quality assurance. These procedures (the whole LC-MS metabolomics analysis) are described in detail in the Zarei et al submitted manuscript that is referred to in the methods section and now also briefly described in the revised version of the manuscript (LINES 597-633).

As the Zarei et al manuscript is not yet available online, for the information of the reviewer, a fuller description of the LC-MS metabolomics data-analytical procedure is included below, and is described in further detail in <https://pubmed.ncbi.nlm.nih.gov/32244411/>

Metabolomics Data Analyses

Data processing was performed separately for each of the four analytical modes. The molecular features were obtained using the MS-DIAL software (Tsugawa *et al.*, 2015). The data were preprocessed using the R software (R Core Team, 2021, <https://www.R-project.org>), where the molecular features were corrected for the drift pattern caused by the LC-MS procedures, and feature quality was assessed based on the quality control samples (Klavus *et al.*, 2020). First, the features were log-transformed. Regularized cubic spline regression was fit separately for each feature on the quality control samples. The smoothing parameter was chosen from an interval between 0.5 and 1.5 using leave-one-out cross validation to prevent overfitting. Features were kept if their non-parametric relative standard deviation and their D-ratio were below 20% and 40%, respectively. In addition, all the features with a relative standard deviation, a non-parametric relative standard deviation, and basic D-ratio below 10% were kept. This additional condition prevents the flagging of features with very low values in all but a few samples. These features tend to have a very high value of non-parametric D-ratio, since the median absolute deviation of the biological samples is not affected by the large concentration in a handful of samples, causing the non-parametric D-ratio to overestimate the significance of random errors in measurements of quality control samples. Thus, other quality metrics were applied with a conservative limit of 0.1 to ensure that only good quality features were kept this way (Klavus *et al.*, 2020; Broadhurst *et al.*, 2018). Missing values were imputed using random forest imputation in two phases. First, only the good quality features were imputed to prevent the flagged features from affecting the imputation. Thereafter, the flagged features were imputed (Stekhoven and Buhlmann, 2011). After imputation, all batches of a single mode were combined, and the median of each quality metric across batches was chosen as an overall quality metric. The preprocessed data was entered in statistical analyses, and molecular features that were statistically significantly affected by the intervention were addressed for metabolite identification following the 4-level metabolite annotation scheme proposed by the Metabolomics Standards Initiative (Sumner *et al.*, 2007).

Compound Identification

The annotation of compounds based on spectral database searches was performed in MS-DIAL. The identification of compounds involved comparing them to purified standards from a library and cross-referencing against several metabolomics databases, including METLIN, MassBank of North America (MoNA), Human Metabolome Database (HMDB), and LIPID MAPS. The MS/MS fragmentation of metabolites was then compared with candidate molecules from the databases and validated using earlier literature on similar compounds. The Metabolomics Center of Biocenter Kuopio also maintains an in-house library of over 1800 authenticated standards, which includes retention time,

	mass-to-charge ratio (m/z), and chromatographic data (including MS/MS spectral data) for all molecules present in the library. Tsugawa et al., 2015: https://doi.org/10.1038/nmeth.3393 Klavus et al., 2020: https://doi.org/10.3390/metabo10040135 Broadhurst et al., 2018: https://doi.org/10.1007/s11306-018-1367-3 Stekhoven and Buhlmann, 2011: https://doi.org/10.1093/bioinformatics/btr597 Sumner et al., 2007: https://doi.org/10.1007/s11306-007-0082-2
3. The author mentioned “The metabolites used in the present analyses are those that were previously found to be affected by the physical activity and dietary intervention from baseline to 8-year-follow-up” in another study. Why only chose previously affected metabolites instead of overall metabolites?	This is a very valid question. The main research objective was to develop a composite exposure score to better understand how exploring the net effect of multiple exposures may differ from investigating the individual exposures separately. Since the aim was focused more on the development, feasibility, and value of a composite exposure score, we used existing clinical data, which could have been measurements of subclinical outcomes such as insulin resistance, blood pressure etc., or metabolomics. Given that metabolomics is more sensitive, it was thought that this would provide a better chance of assessing the main research objective. As such, we moved forward with an already-identified set of metabolites available within the study population, which were those that were found to be affected by the intervention. Once the exposome score has been developed further, it can then be used to investigate associations with the overall set of metabolites and identify metabolomic profiles.
4. For the assessment of air pollution, during the 8 year of follow-up, did authors consider for the movement of study subjects?	Thank you for raising this point. Unfortunately, we did not have data on participant addresses and so it was possible to factor this into our analyses. The concentration of air pollution in Kuopio is quite homogenous across different neighbourhoods, and averaging was used. This is based on unpublished work done by Dr. Mikkonen, who is a coauthor in this study, and the Aerosol Physics Research group at the University of Eastern Finland. They have compared the pollutant concentrations between multiple measurement sites within the city area and found that only by the motorway were the concentrations significantly higher than in other parts of the city, but the particulate pollution or noise were not affecting the exposure levels in residential areas. The air quality network data are not yet published, but there are multiple papers introducing measurements from Puijo tower measurement site, which is close to the city, such as (Portin et al., 2014). They show that the overall levels in Kuopio are low. Thus, if participants remained living in Kuopio, a change-of-address would not have altered the measured/assigned concentrations of their pollution exposure. Portin et al., 2014: https://doi.org/10.5194/acp-14-6021-2014
5. Line 465, In model strategies, how to deal with the potential medical conditions during the study follow up periods? Especially for those related to cardiometabolic conditions, it can also affect the study results.	This is a valid point. However, our study population is a population-based sample of children/adolescents with low prevalence of cardiometabolic diseases. We have assessed the diseases diagnosed by a physician and medications by a questionnaire filled out by the parents at baseline and 2-year follow-up, or adolescents themselves at 8-year follow-up. At baseline, two children in our whole population sample were reported to have some innate cardiac defect (likely mild) which was not specified, but none of the children was reported to have a diagnose of diabetes or cardiometabolic conditions such as high blood pressure or hypercholesterolemia. At 2-year and 8-year follow-up, three subjects had type 1 diabetes but none of the children or adolescents was diagnosed with type 2 diabetes. At 8-year follow-up, 1-2 subjects had diagnoses of high blood pressure or hypercholesterolemia, but they were not using any medication for these conditions at the time of study visits.

	In terms of cardiometabolic conditions, in this study we were interested in looking at associations of the exposome score with serum metabolites measured by LC-MS and NMR related to cardiometabolic risk factors. If cardiometabolic conditions that developed during the study were controlled for, it is possible that we would simply be adjusting away the associations/effects of the exposures measured in this study. Moreover, the number of children with diagnosed cardiometabolic conditions was very low. This is an interesting point though, and future studies that explore the exposome score further will assess it in adult populations using a less-variable outcome than metabolomics data (such as disease incidence).
6. In statistical analysis, I would suggest conduct multiple comparison correction for all results as to reduce the false positive discovery induced by large metabolomics data.	Thank you for the suggestion. The p-values presented in the manuscript are unadjusted p-values. Due to the low concentrations of pollutants, and the large number of tests associated with metabolomics studies, it was considered that the effect sizes of the pollutants would be quite weak and presenting adjusted p-values would run the risk of completely diluting significance. The supplementary material has been updated to include adjusted p-values.
7. For table1, suggest to add one column of p-value for chi-square or ANOVA test between three groups. Also, add unit for cotinine concentrations.	Thank you for your suggestion. We have added a p -value column to Table 1 (PAGE 47). In this study, cotinine is a metabolite measured using LC-MS, and thus does not have a typical measurement unit, such as ng/mL.
8. For table 2, what are metabolite abundances? Are these metabolite intensities? Suggest to clarify.	Thank you for your question. The used metabolite values are the peak area abundances resulting from the computational data matrix generation in non-targeted metabolomics, and thus they are reflecting the intensity/concentration of the metabolite in the sample set when compared to the other measured samples. This procedure is typical for non-targeted metabolomics analysis, and detail on it can be found in e.g., our article describing all the parts of the analysis: https://pubmed.ncbi.nlm.nih.gov/32244411/
9. For all supplement materials, they can't be shown completely in pdf file. Need to remake new one.	Thank you for pointing this out. It is unfortunate that the information in the supplemental table could not be easily viewed. The manuscript was submitted using a "transfer service", and it appears there were issues with the automatic formatting of the supplementary material. The information was originally provided in an excel file, and seemingly was converted to PDF format.
10. For figure2, why some y-axis showed metabolite abundances and some showed concentration? Suggest to clarify. Also, panel B seems decrease over time and panel E seems increase, which is opposed to the statement in note. Please double check.	Thank you for your question. This study used two methods to measure metabolites – LC-MS and NMR. The use of NMR allows for direct quantitation of metabolites and thus can measure the circulating concentrations, hence the metabolites measured by NMR are reported with concentrations. Metabolites measured by LC-MS are measured with their arbitrary peak area, and the values across the measured sample series allow comparison on the relative intensities of the peaks, i.e. the levels of the compounds and thus the values are presented as "abundances", which is common when referring to metabolites measured in non-targeted LC-MS metabolomics. We have now amended the figure description to clarify which metabolites were measured by LC-MS, and which metabolites were measured by NMR (LINES 1051-1053). "This figure illustrates the different patterns of how adiposity, measured by BMI-SDS, modified the associations of the exposome score with serum metabolites measured by NMR (panels A, B, C, D) and LC-MS (panels E, F)." Thank you for highlighting the point on the panels. We have used a different example metabolite which provides a clearer visual that matches with the statement.
11. Overall, the results lack the comparison of significant	Thank you for this suggestion. We agree that the overlap in results can benefit from visualisation. This has been included for the additional sensitivity analyses

metabolites between three different models (original, mediator and modification) for LCMS and NMR, respectively. Suggest to visualize them (eg: Venn diagram)

we have completed (**Supplementary Figures S1-S4**). However, after completing a similar visualization of the overlap between the original, mediation and modification results, we are uncertain of the added value this visualization will provide. This is because the mediator models cannot have results independent of the original model (and thus, there can be no overlap between mediator and modifier independent of the original model). We have attached the figures here for your viewing, and if the reviewer/editor still believes they will provide added value to the reader, then we are happy to accommodate this suggestion.

Figure 1. Comparison of significant associations of original (LCMS), mediation and moderation models with serum metabolites measured by LC-MS

Figure 2. Comparison of significant associations of original (NMR), mediation and moderation models with serum metabolites measured by NMR

12. Line 209, the author mentioned: “many of these metabolites observed to be decreased in individuals with a higher exposome score, indicative of a higher overall exposure to harmful environmental and lifestyle factors.” Does this mean that lower levels of these metabolites indicate “bad”? It’s not always

Thank you for raising this point. Lower (or higher) levels of metabolites (or other biochemical measurements) does not always mean it is “bad”, or that these changes will lead to adverse effects on health. In many cases, clinical variables, such as cholesterol, have optimal ranges, and it is outside these ranges that we often see increased risks of negative health outcomes. This is true in the example you provided – there is a balance between pro- and anti-inflammatory compounds to ensure the human body is capable of effectively responding to stressors. Inflammatory responses are necessary e.g., to ingest and eliminate apoptotic cells and pathogens. Adverse effects occur when there is a prolonged shift towards, for example, a pro-inflammatory state, with chronic inflammation implicated in the development of cardiometabolic diseases (Donath, Meier and Boni-Schnetzler, 2019; Aksentijevich *et al.*, 2020)

the case for pro-inflammatory metabolites.	Ishida et al., 2013: https://doi.org/10.1016/j.ejphar.2012.11.030 Xu et al., 2016: https://doi.org/10.1097/FJC.0000000000000413 Ziegler et al., 2020: https://doi.org/10.1016/j.redox.2020.101581 Donath, Meier and Boni-Schnetzler, 2019: https://doi.org/10.1210/er.2019-00002 Aksentijevich et al., 2020: https://doi.org/10.1016/j.tcm.2019.11.001
13. Can author also briefly discuss the discrepancy of the direction of associations between LCMS and NMR? For example, line 209-211; line 226-228.	Thank you for your question. While opposing direction of associations are observed, these are not surprising based on the fact that the metabolite(s) being measured are different. Based on the literature, some of the significant compounds measured by LC-MS observed in our study demonstrate anti-inflammatory effects. The only compound measured by NMR that went in the same direction, which was a negative association, was acetate, a compound that is also suggested to have anti-inflammatory effects. On the other hand, positive associations were observed with many metabolites measured by NMR – these were with compounds that have been positively associated with adverse cardiometabolic outcomes, such as measures of triglycerides and cholesterol and certain amino acids. As such, while the direction of the associations may be opposing, the biological interpretation is quite consistent, that is, decreases in compounds that appear to confer protective effects on cardiometabolic health, and increases in compounds that appear to be associated with adverse cardiometabolic outcomes.
14. Suggest to briefly mention why LCMS and NMR these two methods are applied? Advantages or disadvantages, and how they complement with each other?	Thank you for this suggestion. While we originally included a reference to a paper which discusses the advantages and disadvantages and complementary nature of the two methodologies in the Introduction (Marshall and Powers, 2017), we have added a brief overview of these aspects to the introduction and included further references to improve the overall clarity and highlight the benefits of this dual-method approach (LINES 107-110). “LC-MS is highly sensitive and can detect a large range of metabolites in very low levels in biological samples. In contrast, NMR can measure larger compounds e.g., lipoprotein particles that are not amenable for analysis in general LC-MS procedures.”^{39,40} Marshall and Powers, 2017: https://doi.org/10.1016/j.pnmrs.2017.01.001
15. As mentioned in limitation, I’m still concerned about the performance of scoring approach as this may oversimplify the complexities of these exposures which are non-linear and have interactive relationships between each other. For example, air pollution can affect physical activity, SES can affect diet choice etc. These are more like to be synergistic. Perhaps consider some sensitivity analysis, like leave out analysis-remove one exposure at a time.	This is an extremely valid point, and we agree with the notion that these exposures are likely to have interactions with each other. The ability to accurately measure the effect of multiple exposures is a major challenge in exposomics research and was the goal of a recent “exposome data challenge event” (Maitre et al., 2022). Often in biomedical and environmental research, many statistical approaches look at single exposures, such as single-pollutant models within the context of air pollution. Even if attempts are made to quantify the effect of multiple exposures, through some form of mixture analysis with e.g., quantile g-computation, these generalized additive models often still investigate associations without incorporating other exposures such as diet, physical activity, socioeconomic status etc. The composite score developed in this paper is not intended as an “end product” that is capable of accurately catching the complex nature of the impact of various environmental and lifestyle exposures. Instead, it was developed to explore the differences in significant findings when investigating a single exposure category (such as diet or air pollution) compared to assessing the associations of combined exposures, and to highlight that while significant associations may exist when assessing individual exposures, these associations may “disappear” when investigating

the net impact of combined exposures, while there may also be a sort of cumulative effect where the combination of multiple exposures results in a significant association, while the individual exposures may not be significantly associated with the outcome.

The leave-one-out analysis is a helpful suggestion. It will highlight if there are specific “exposure categories” that are more heavily influencing the results and further support the notion of conducting research which aims to investigate the net impact of multiple exposures on an individual’s overall health. In the original submission, we contrast the associations of the individual exposure categories with the composite exposome score. To expand upon this further, we have also added the suggested analysis (Results: LINES 286-303] and elaborated upon it in the discussion [LINE 513-526]. See also Supplementary Tables S7-S8 and Supplementary Figures S1-S2.

Results:

“Leave-one-out analysis

The original exposome score model and leave-out models were associated with 54 serum metabolites measured by LC-MS (Supplementary Table S7). The directions of the significant associations of the exposome score with serum metabolites measured by LC-MS were consistent across the various leave-out models (Supplementary Table S7). Most of the models were observed to have a similar number of significant associations with metabolites as the original exposome score model, ranging between 13 and 16 (Supplementary Figure S1). The deviation to this trend was found with the removal of the air pollution category from the exposome score; this model had 29 significant associations with metabolites, which was the largest number of such associations (Supplementary Figure S1, Supplementary Table S7).

The original exposome score model and leave-out models were associated with 27 serum metabolites measured by NMR (Supplementary Table S8). The directions of the significant associations of the exposome score with serum metabolites measured by NMR were consistent across the various leave-out models (Supplementary Table S8). In comparison with the original exposome score model, the leave-out models for diet, activity, sleep, and socioeconomic status were all significantly associated with less metabolites (8, 5, 12 and 8, respectively). The exception to this was the leave-out-model for pollution, which was significantly associated with 23 metabolites (Supplementary Figure S2, Supplementary Table S8).”

Discussion:

“To assess the contribution of the individual exposure components to the exposome score, we conducted a leave-one-out sensitivity analysis. No individual component appeared to be influencing the directions of the associations, with the direction remaining consistent between the exposome score and the various leave-out models. However, it appeared that the air pollution score had a large influence on the number of significant associations, with the leave-out model for pollution significantly associated with the largest number of metabolites measured by both LC-MS and NMR. Interestingly, when we inspect metabolites that were significantly associated with the pollution score and at least one other exposure component, we can see that the directions differ in most instances. It is possible that the pollution score is exerting a small effect in the opposing direction to the other exposures, and while not a large enough effect to be significant, it may be enough to influence the overall exposome score, as evidence by the comparison of the original exposome score with the exposome score from which the air pollution component was removed.”

Further to your point about concerns regarding the use of scores, we have also completed a further sensitivity analysis by testing different scoring methods for diet quality, to show that different approaches to measure diet quality (being just one of five exposure categories) can quite dramatically change the findings. These have been presented in LINES 273-285 and discussed in LINE 487-512.

Results:

“Comparison of diet scores within exposome score

The use of the Finnish Children Healthy Eating Index, Mediterranean Diet Score, or Baltic Sea Diet Score in the formulation of the exposome score was significantly associated with 16, 27 and 20, respectively, serum metabolites measured by LC-MS (Supplementary Table S3). Altogether, 11 metabolites were common to all three models. However, there were four, eight and two unique associations when the Finnish Children Healthy Eating Index, the Mediterranean Diet Score, and the Baltic Sea Diet Score were used in the exposome score, respectively (Supplementary Figure S3).

The use of the Finnish Children Healthy Eating Index, the Mediterranean Diet Score, or the Baltic Sea Diet Score in the formulation of the exposome score was significantly associated with 15, 11 and 22, respectively, serum metabolites measured by NMR (Supplementary Table S4). Four metabolites were common to all three models. However, there were five, two and seven unique associations when the Finnish Children Healthy Eating Index, the Mediterranean Diet Score, and the Baltic Sea Diet Score were used in the exposome score, respectively (Supplementary Figure S4).”

Discussion:

“To demonstrate the challenges of developing a composite exposure score and exploring its feasibility, we conducted a sensitivity analysis using just one component of the exposome score (diet), where the use of different, validated, approaches to assess diet quality were compared. The original exposome score model, using the Finnish Children Healthy Eating Index, was compared to two sensitivity exposome models using either the Baltic Sea Diet Score or the Mediterranean Diet Score. While there was some overlap between the three models, there were also a number of unique associations with serum metabolites measured by LC-MS and NMR based on the diet score incorporate into the exposome score (Supplementary Figures S3-S4). Interestingly, when we compared the results of the sensitivity analyses using the Finnish Children Healthy Eating Index and the Baltic Sea Diet Score, both of which have been validated in a Finnish population,^{20,21} we found that almost a third of the metabolites measured by LC-MS and NMR that were associated with the original exposome score were not found to be related to the exposome score with the Baltic Sea Diet Score. Future research should consider the suitability of selected measures to the population studied, and further research is necessary to determine optimal approaches to assess the impact of diet on an individuals’ overall exposome, whether this be through a diet quality approach, or the assessment of specific foods or nutrients. Future research should also explore the impact of ultra-processed foods, which have been positively associated with obesity and other cardiometabolic risk factors in children and adolescents.¹⁰⁵ Specifically, animal-based products and sugar-sweetened beverages should be examined, which were recently determined as the main subgroups of ultra-processed foods associated with cardiometabolic multimorbidity in adults across 10 European countries.¹⁰⁶ These are insights based on the diet component of the exposome score, with special care likely needed for all components incorporated into a composite exposure score. This highlights the need to carefully reflect upon and select appropriate measures, and thus researchers should be careful in the planning of composite scores to measure

	multiple exposures, and the approaches used to assess the components to be incorporated.” At this point, it is probably too much information for this specific manuscript and better suited to a follow-up project on further development of a composite exposure measure, however we also investigated the interactions between the individual exposure components. For the purpose of this, we chose a single outcome (Glycoprotein acetyls), instead of exploring each interaction for each metabolite. Glycoprotein acetyls have been suggested as a biomarker of early cardiovascular risk as early as adolescence (Chiesa et al., 2022). We found that every single combination of diet_score, sleep_score, pollution_score, socioeconomic_score, activity_score was observed to significantly interact, and the directions of these interactions differed depending on the combination. Maitre et al., 2022: https://doi.org/10.1016/j.envint.2022.107422 Chiesa et al., 2022: https://doi.org/10.1161/JAHA.121.024380
16. One more point in limitation, the large percentage of loss to follow up during periods can introduce bias to study.	Thank you for highlighting this, the suggestion has been added to the limitations section [LINE 557-558]. “It should also be noted that there was a large number of dropouts, around 45% of the number of participants at baseline, which may introduce bias to the study. “
17. Line 38, suggest to add SES in these factors as the authors also considered SES in analysis.	Thank you for noting this, it is a valid suggestion to specify each of the exposures to be integrated into the composite score. It was not the intention to list every variable that was incorporated into the composite score; for example, we have also not mentioned sleep in this sentence. The references included at the end of the statement relate to the factors that have been listed (air pollution, diet, physical activity). The purpose of this sentence was just to highlight that we are exposed to many aspects throughout our lives, both environmental and personal, that have an impact on our health.
18. Line 40-42, “However, environmental and lifestyle exposures do not exert their physiological effects in isolation. Instead, there is complex interplay between external exposures, and with their associated internal physiological responses.” Needs reference to support this statement.	Thank you for the suggestion. Multiple references have now been provided [LINE 46; references 7-11], for potential interactions between external exposures (such as between pollutants), as well as evidence supporting lifestyle-environmental interactions and impact on health.
19. Line 378, the overall study design seems like a cohort study instead of clinical trial in current analysis. What’s the intervention for intervention group? Is intervention/ control group related to current study? If not, suggest to delete.	The original study is that of a non-randomised intervention. The intervention group received individualised and family-based physical activity and dietary counselling sessions. In the context of this secondary analysis, all participants were analysed as a single cohort. Per the suggestion, the specific information on the intervention and control groups has been removed (LINES 589-593).
20. Line 394, suggest to change “second-hand smoking” to “environmental exposure to smoking” as third-hand smoking can also be an underlying source for cotinine.	Thank you for the suggestion. The sentence has been amended per your suggestion (LINES 634-635). “Serum cotinine, a nicotine metabolite measured by LC-MS, was used as a biomarker of environmental exposure to smoking. “

REVIEWERS' COMMENTS:

Reviewer #1 (Remarks to the Author):

I have no further comments.

Reviewer #2 (Remarks to the Author):

I pleased to acknowledge that the authors have diligently addressed all the comments and suggestions provided during the review process. Their efforts in revising the manuscript have significantly improved the overall quality of the paper. I believe that the revisions have enhanced the clarity, coherence, and significance of the research findings. Therefore, I recommend accepting their publication. Thank you for the opportunity to review this.

Reviewer #3 (Remarks to the Author):

The questions and suggestions proposed are well addressed in response letter and manuscript. The authors conducted more comprehensive sensitivity analysis to valid the study results. Overall, this study is novel and quite clinically meaningful. The development of "exposome score" supports the future study on exposome field.